# Developing a multivariate prediction model of antibody features associated with protection of malaria-infected pregnant women from placental malaria

Elizabeth H Aitken[1†], Timon Damelang[2†], Amaya Ortega-Pajares[1†], Agersew Alemu[1], Wina Hasang[1], Saber Dini[3], Holger W Unger[1,4,5], Maria Ome-Kaius[6], Morten A Nielsen[7], Ali Salanti[7,8], Joe Smith[9,10], Stephen Kent[2], P Mark Hogarth[9,11,12,13], Bruce D Wines[11,12,13], Julie A Simpson[3], Amy W Chung[2‡], Stephen J Rogerson[1‡*]

[1]Department of Medicine, University of Melbourne, the Doherty Institute, Melbourne, Australia; [2]Department of Microbiology and Immunology, University of Melbourne, the Doherty Institute, Melbourne, Australia; [3]Centre for Epidemiology and Biostatistics, Melbourne School of Population and Global Health, University of Melbourne, Melbourne, Australia; [4]Department of Obstetrics and Gynaecology, Royal Darwin Hospital, Darwin, Australia; [5]Menzies School of Health Research, Darwin, Australia; [6]Walter and Eliza Hall Institute of Medical Research, Parkville, Australia; [7]Centre for Medical Parasitology, Department of Microbiology and immunology, University of Copenhagen, Copenhagen, Denmark; [8]Department of Infectious Disease, Copenhagen University Hospital, Copenhagen, Denmark; [9]Seattle Children's Research Institute, Seattle, United States; [10]Department of Pediatrics, University of Washington, Seattle, United States; [11]Immune Therapies Group, Centre for Biomedical Research, Burnet Institute, Melbourne, Australia; [12]Department of Clinical Pathology, University of Melbourne, Melbourne, Australia; [13]Department of Immunology and Pathology, Monash University, Melbourne, Australia

*For correspondence: sroger@unimelb.edu.au

[†]These authors contributed equally to this work
[‡]These authors also contributed equally to this work

Competing interests: The authors declare that no competing interests exist.

## Abstract

**Background:** *Plasmodium falciparum* causes placental malaria, which results in adverse outcomes for mother and child. *P. falciparum*-infected erythrocytes that express the parasite protein VAR2CSA on their surface can bind to placental chondroitin sulfate A. It has been hypothesized that naturally acquired antibodies towards VAR2CSA protect against placental infection, but it has proven difficult to identify robust antibody correlates of protection from disease. The objective of this study was to develop a prediction model using antibody features that could identify women protected from placental malaria.

**Methods:** We used a systems serology approach with elastic net-regularized logistic regression, partial least squares discriminant analysis, and a case-control study design to identify naturally acquired antibody features mid-pregnancy that were associated with protection from placental malaria at delivery in a cohort of 77 pregnant women from Madang, Papua New Guinea.

**Results:** The machine learning techniques selected 6 out of 169 measured antibody features towards VAR2CSA that could predict (with 86% accuracy) whether a woman would subsequently have active placental malaria infection at delivery. Selected features included previously described associations with inhibition of placental binding and/or opsonic phagocytosis of infected

erythrocytes, and network analysis indicated that there are not one but multiple pathways to protection from placental malaria.

**Conclusions:** We have identified candidate antibody features that could accurately identify malaria-infected women as protected from placental infection. It is likely that there are multiple pathways to protection against placental malaria.

**Funding:** This study was supported by the National Health and Medical Research Council (Nos. APP1143946, GNT1145303, APP1092789, APP1140509, and APP1104975).

## Introduction

The burden of *Plasmodium falciparum* malaria is greatest in young children and pregnant women (*World Health Organisation, 2019*). Among women with lifelong malaria exposure, pregnant women have increased susceptibility to malaria, and complications include maternal mortality, severe maternal anemia, miscarriage and stillbirth, and impaired fetal growth or prematurity leading to low birth weight delivery (*Steketee et al., 2001*; *Guyatt and Snow, 2001*; *Desai et al., 2007*; *Moore et al., 2017a*; *Moore et al., 2017b*). The prevalence, intensity, and consequences of infection decline with increasing gravidity (*Guyatt and Snow, 2001*).

The susceptibility of pregnant women to *P. falciparum* malaria is in part due to the ability of infected erythrocytes (IEs) to sequester in the maternal blood spaces of the placenta (*Rogerson et al., 2007*), where IEs that express VAR2CSA adhere to chondroitin sulfate A (CSA), a glycosaminoglycan chain on syndecan-1 expressed by the placental syncytiotrophoblast (*Salanti et al., 2004*; *Ayres Pereira et al., 2016*). VAR2CSA is a unique 350 kDa protein that is a member of the *P. falciparum* erythrocyte membrane protein 1 (*Pf*EMP1) family of variant surface antigens (VSA). *Pf*EMP1s are encoded by *var* genes and are composed of multiple Duffy binding-like (DBL) domains and cysteine-rich interdomain regions (CIDR). VAR2CSA has six DBL domains, with CSA adhesion being strongly associated with the DBL2 domain, which is present in the N-terminal region of the protein (*Doritchamou et al., 2013*). Two constructs based on this region have entered clinical trials as vaccine candidates to prevent placental malaria (*Mordmüller et al., 2019*; *Sirima et al., 2020*).

Antibodies to VAR2CSA, in its native form on IE or expressed as recombinant protein, develop in a parity-dependent manner, block adhesion to CSA, and opsonize IE for phagocytosis by monocytic cells (*Fried et al., 1998*; *Ricke et al., 2000*; *Salanti et al., 2004*; *Keen et al., 2007*). Some studies report associations between measures of single antibody features and protection from malaria complications such as placental infection, impaired fetal growth, prematurity, or anemia (*Duffy and Fried, 2003*; *Staalsoe et al., 2004*; *Feng et al., 2009*). However, identifying targets and features of protective antibody responses to malaria is challenging because exposure to infection is accompanied by the production of antibody to many parasite antigens, only a fraction of which contributes to protective immunity (*Crompton et al., 2010*). The large size and multidomain structure of VAR2CSA adds complexity to the identification of targets and features of protective immune response. In a recent systematic review, antibody responses to VAR2CSA were often correlated with exposure to placental malaria, rather than protection from infection (*Cutts et al., 2020*), although some studies indicate an association between antibody to IE and protection from low birth weight in subsets of pregnant women (*Duffy and Fried, 2003*; *Staalsoe et al., 2004*; *Feng et al., 2009*).

To identify antibody responses that protect women from placental malaria, an alternative approach that examines multiple antibody features and controls for exposure may be needed. Placental malaria mostly occurs when antigenic variation (*Roberts et al., 1992*) leads to a parasite that has infected a pregnant woman, expressing VAR2CSA. These parasites can flourish by sequestering in the placenta in women who lack VAR2CSA-specific immunity (*Hviid, 2004*). By contrast, some pregnant women with peripheral blood infection remain free of placental malaria, and we postulate that these women have antibody responses that specifically protect them against VAR2CSA and placental malaria.

Antibody measures towards the IE have traditionally focused on the quantity of bound antibodies and their ability to inhibit IE binding to CSA (*Cutts et al., 2020*), but there is increasing interest in how antibodies that recognize IE engage with innate immune cells and activate complement (*Aitken et al., 2020*), which are both determined by biophysical features of the Fab and Fc regions

of antibody (reviewed in *Arnold and Chung, 2018*). In recent years, detailed functional and biophysical characterization of antibody responses has led to the identification of specific antibody determinants that correlate with vaccine-induced protection from HIV (*Chung et al., 2015*), control of latent tuberculosis infection (*Lu et al., 2016*), transplacental transfer of antibody (*Martinez et al., 2019*; *Jennewein et al., 2019*), and correlates of vaccine-induced protection in human malaria challenge models (*Suscovich et al., 2020*). This approach, which has been termed systems serology, is well suited to placental malaria because there is a clear relationship between a single antigen (VAR2CSA) and a specific pathology (the sequestration of parasites in the placenta, reviewed in *Ataíde et al., 2014*).

The objective of this study was develop a prognostic model using antibody features to identify women protected from placental malaria. In this prospective study, we identified features, functions, and targets of naturally acquired antibody to VAR2CSA that contribute to protection against placental malaria. Pregnant Papua New Guinean women who were participating in a trial of intermittent preventive treatment against malaria (*Unger et al., 2015*) were categorized at delivery as currently uninfected, having placental malaria, or having a non-placental infection. Using plasma samples collected at enrollment in mid-pregnancy, we performed extensive profiling of antibody responses to VAR2CSA recombinant proteins and IEs expressing VAR2CSA. Machine learning techniques revealed that 6 of 169 antibody features measured mid-pregnancy were able to discriminate between women who had placental malaria and non-placental infection at delivery with high accuracy. Five of the six leading antibody features we found were related to inhibition of placental sequestration and/or opsonization for phagocytic clearance.

## Materials and methods

### Experimental model and subject details

#### Human subjects

Study participants were recruited between November 2009 and August 2012 as part of a randomized controlled trial of Intermittent Preventive Treatment in Pregnancy (IPTp) (*Unger et al., 2015*) (ClinicalTrials.gov NCT01136850), in which pregnant women received either three courses of sulfadoxine pyrimethamine (SP) and azithromycin or one course of SP and chloroquine. Women were recruited at 14–26 gestation weeks and followed up until delivery. All women presenting for their first antenatal visit at one of the nine participating health centers in Madang Papua New Guinea were invited to participate in the original cohort study. Exclusion criteria included a gestation of >26 weeks, hemoglobin <6 g/dl with symptomatic anemia, previous serious adverse reactions to the IPTp study medications, permanent disability or chronic medical conditions, known multiple pregnancy, age <16 years, or known unavailability to follow-up *Unger et al., 2015*. Demographic data collected at enrollment included maternal gravidity, age, residence, and bed net use. Peripheral blood collected at 14–26 gestation weeks, before initiation of IPTp, was used in the antibody assays. Peripheral blood collected at delivery was used to prepare blood smears for malaria microscopy and to extract DNA for qPCR for malaria parasites (*Unger et al., 2015*). Placental biopsies were formalin fixed and paraffin embedded. Giemsa stained sections of placental biopsies were examined for malaria infection (*Lufele et al., 2017*). Women were selected for inclusion in this case-control study based on the presence of *P. falciparum* IEs in peripheral blood and/or placenta at delivery. Groups included women with no evidence of infection in peripheral blood by PCR or light microscopy nor on examination of placental histology at delivery (n = 50); women with active placental malaria characterized by detection of *P. falciparum* IEs by placental histology (n = 50); and women with *P. falciparum* infection by light microscopy and/or PCR in the peripheral blood but with no IEs detected by placental histology examination (n = 27). Sample size was based on published work that used a similar approach to investigate antibody responses to infectious diseases (*Lu et al., 2016*); in addition, sample size for the non-placental infection group was also limited by sample availability as few women met the criteria for inclusion. All clinical data were collected, and grouping of women by outcome was completed prior to measurements of antibody features included in the model. The three groups were frequency matched for primigravidity, IPTp regime receipt, bed net use, rural residency, and age. See *Table 1* for details on clinical characteristics of participants.

**Table 1.** Clinical characteristics of the three groups of pregnant women at the time of antibody feature measurement at enrollment (14–26 weeks' gestation) and also at delivery.

| | Non-infected at delivery | | Placental malaria at delivery | | Non-placental infection at delivery | |
|---|---|---|---|---|---|---|
| | N = 50 | | N = 50 | | N = 27 | |
| **Enrollment** | | | | | | |
| Mean age (years), SD | 24.5 | 5.3 | 24.0 | 5.0 | 23.1 | 4.4 |
| *Residence*, N (%) | | | | | | |
| Rural | 37 | (74.0) | 38 | (76.0) | 18 | (66.7) |
| Non-rural | 13 | (26.0) | 12 | (24.0) | 9 | (33.3) |
| *Ethnicity*, N (%) | | | | | | |
| Sepik | 6 | (12.0) | 11 | (22.0) | 3 | (11.1) |
| Madang/Morobe | 39 | (78.0) | 30 | (60.0) | 22 | (81.5) |
| Highlander | 3 | (6.0) | 5 | (10.0) | 1 | (3.7) |
| Other | 2 | (4.0) | 4 | (8.0) | 1 | (3.7) |
| Formal schooling, N (%) | 46 | (92.0) | 46 | (92.0) | 25 | (92.6) |
| Smoking, N (%) | 9 | (18.0) | 11 | (22.0) | 6 | (22.2) |
| Betel nut user, N (%)† | 41 | (82.0) | 41 | (82.0) | 24 | (88.9) |
| Alcohol, N (%) | 2 | (4.0) | 2 | (4.0) | 2 | (7.4) |
| *Gravidity*, N (%) | | | | | | |
| Primigravidae | 26 | (52.0) | 29 | (58.0) | 14 | (51.9) |
| Secundigravidae | 8 | (16.0) | 7 | (14.0) | 8 | (29.6) |
| Multigravidae | 16 | (32.0) | 14 | (28.0) | 5 | (18.5) |
| *IPTp regime*, N (%) | | | | | | |
| SPCQ | 27 | (54.0) | 30 | (60.0) | 18 | (66.7) |
| SPAZ | 23 | (46.0) | 20 | (40.0) | 9 | (33.3) |
| Mean gestational age (days), SD | 145.9 | 31.4 | 147.5 | 31.3 | 152.2 | 19.8 |
| Mean maternal weight (kg), SD* | 54.7 | 13.1 | 53.5 | 8.2 | 54.1 | 7.4 |
| Mean maternal height (cm), SD† | 154.3 | 5.9 | 154.6 | 6.9 | 154.4 | 6.0 |
| Bed net use, N (%) | 34 | (68.0) | 40 | (80.0) | 21 | (77.8) |
| Hb (g/dL), mean SD‡ | 9.7 | 1.2 | 9.3 | 2.0 | 9.4 | 1.2 |
| Light microscopy positive for Pf, N (%) | 2 | (4.0) | 4 | (8.0) | 4 | (14.8) |
| PCR positive for Pf, N (%) | 4 | (8.0) | 5 | (10.0) | 5 | (18.5) |
| *Delivery* | | | | | | |
| Placenta light microscopy positive for Pf, N (%) | 0 | (0) | 50 | (100) | 0 | (0) |
| Peripheral blood light microscopy positive for Pf, N (%) | 0 | (0) | 10 | (20) | 13 | (48.2) |
| Peripheral blood PCR positive for Pf, N (%) | 0 | (0) | 13 | (26) | 21 | (77.8) |
| Placental blood PCR positive for Pf, N (%)§ | 1 | (2.0) | 10 | (20) | 10 | (37.04) |
| Birthweight (g), SD | 3062 | 546 | 2827 | 501 | 2840 | 416 |
| Gestation at delivery (days), SD¶ | 278 | 18 | 279 | 16 | 280 | 13 |
| Mean Hb (g/dL), SD | 10.1 | 2 | 9.5 | 1.9 | 10.2 | 1.4 |

* One participant with missing data on betel nut use in placental malaria.

†One participant with missing data on weight in the non-infected group.

‡ Missing Hb data, five in non-infected group, four in placental malaria group, and two in the non-placental infection group.

§ Missing placental PCR data in five non-infected, eight placental malaria, and four non-placental infection women.

¶One participant with missing data on gestation length at delivery in the non-infected group.

SD: standard deviation; Hb: hemoglobin; PCR: polymerase chain reaction; IPTp: intermittent preventive treatment in pregnancy; SPAZ: sulfadoxine pyrimethamine-azithromycin; SPCQ: sulfadoxine pyrimethamine-chloroquine.

## Parasite cell lines

*P. falciparum* IE of the parasite lines CS2 and 3D7 were cultured as previously described (*Chandrasiri et al., 2014*). Cultures were synchronized as needed by sorbitol lysis (*Lambros and Vanderberg, 1979*) and IE were regularly selected for expression of knobs by gelatin flotation (*Goodyer et al., 1994*). Cell cultures were mycoplasma negative (tested for mycoplasma using the MycoAlert kit [Lonza, Mount Waverley, Australia] as per manufacturer's instructions). For the binding inhibition assays, the parent lines of CS2 and 3D7 (FCR3 and NF54, respectively) were used, cultured, and selected for CSA adhesion as described previously (*Nielsen and Salanti, 2015*).

## Monocyte cell line THP-1

THP-1 cells were mycoplasma negative and cultured as previously described (*Ataíde et al., 2010*).

## Primary leukocytes

Neutrophils were isolated from fresh venous blood collected in lithium heparin vacutainers (BD, Scoresby, Australia) using the EasySep Direct Human Neutrophil Isolation Kit (STEMCELL Technologies, Tullamarine, Australia) as per manufacturer's instructions. Neutrophil purity was assessed by cell morphology using light microscopy of Giemsa stained smears of the isolated cells, and viability was assessed using trypan blue exclusion. Monocytes were isolated from both fresh venous blood, collected in lithium heparin vacutainers (BD), as well as from buffy coats supplied from the Australian Red Cross Blood Service. Monocytes were isolated by negative selection using the RosetteSep Human Monocyte Enrichment Cocktail (STEMCELL Technologies) as per manufacturer's instructions. Monocyte purity was assessed by staining (anti-CD14 antibody, BioLegend, San Diego, CA) and measuring CD14$^+$ cells by flow cytometry. Monocytes were either frozen in fetal bovine serum (FBS) in 20% dimethyl sulfoxide (DMSO) in liquid nitrogen for later use or used immediately after isolation. Natural killer (NK) cells were isolated from fresh venous blood collected in sodium heparin vacutainers (BD). NK cells were isolated by negative selection using the RosetteSep Human NK Enrichment Cocktail (STEMCELL Technologies) as per manufacturer's instructions.

## Method details

### Samples and controls

Plasma samples of study participants were obtained from venous blood collected into sodium heparin vacutainers upon enrollment into the trial at 14–26 weeks' gestation and plasma was separated by centrifugation and stored at −80°C until used. To generate a positive control, plasma samples from pregnant women in Malawi with high levels of IgG that recognizes IEs of the *P. falciparum* line CS2 (a VAR2CSA-expressing and CSA-binding line; *Elliott et al., 2005b*) were pooled. Negative controls were sera from individual Melbourne donors obtained from the Australian Red Cross Blood Service or plasma from Melbourne donors collected in lithium heparin vacutainers (BD). All controls were kept at −80°C until use. Antibody features were measured after grouping of women based on outcome. Researchers measuring antibody features were blinded to outcome data of individual samples; this was done by allocating all samples a number (which was not associated with clinical outcomes and which was different from the original cohort study number), samples were decoded after acquisition of the antibody feature was complete. For some experiments, IgG was purified from plasma and serum using Melon Gel purification kits (Thermo Fisher Scientific, Scoresby, Australia) as per manufacturer's instructions. Purified IgG was quantified before use with a human IgG ELISA development kit (MabTech, Preston, Australia) as per manufacturer's instructions.

## VAR2CSA DBL recombinant antigens

Twelve recombinant proteins consisting of subunits or DBL domains of VAR2CSA were used in experiments (see *Supplementary file 1* for a table detailing proteins used). For some experiments, proteins were biotinylated using EZ-link Sulfo NHS-LC-Biotin kit (Thermo Fisher Scientific) as per manufacturer's instructions.

## Infected erythrocytes

The CSA-binding phenotypes of CS2 and 3D7 IEs were monitored by measuring binding of trophozoite stage IEs in static binding assays to recombinant human CD36 (R&D systems, Noble Park,

Australia) and CSA (Sigma-Aldrich, Macquarie Park, Australia) as previously described (*Yosaatmadja et al., 2008*). 3D7 was selected for a CSA-binding phenotype by flow cytometry. Purified trophozoite stage 3D7 IEs were incubated with fluorescein isothiocyanate (FITC)-labeled CSA 100 µg/mL (Creative PEGWorks, Durham, NC) for 30 min at 37°C. Purified IE incubated with CSA-FITC and 3D7 without CSA-FITC were used to set the gates. IEs were then gated based on forward and side scatter and sorted based on FITC. For some assays, trophozoite stage IE were purified by Percoll gradient and stained with dihydroethidium (DHE) 25 µg/mL (Sigma-Aldrich). For the binding inhibition assays, FCR3 and NF54 IE were selected for a CSA-binding phenotype by panning on BeWo cells as previously described (*Nielsen and Salanti, 2015*).

## Antibody features to VAR2CSA by multiplex

Multiplex assays were used to detect plasma levels of antigen-specific antibodies (Abs) towards VAR2CSA DBL domains. Features assessed included total IgG, IgG1, IgG2, IgG3, IgG4, IgA1, IgA2, IgM as well as antibody engagement with complement (C1q) and FcγRs (FcγRI, FcγRIIa, FcγRIIIa, and FcγRIIIb). To conduct the multiplex assays, DBL domains (*Supplementary file 1*) of the VAR2CSA protein were coupled to Bio-Plex magnetic carboxylated microspheres (Bio-Rad) as per manufacturer's instructions, blocked with PBS plus 0.1% bovine serum albumin (BSA), then stored in PBS-0.05% sodium azide at $-80°C$.

On the day of the assay, the DBL-coupled microspheres were resuspended at $1 \times 10^4$ beads/mL in 1:100 dilution of plasma in PBS-1% BSA and 50 µL were aliquoted into each well of a 96-well round bottom plate (Greiner Bio-One, Kremsmünster, Austria). Plates were incubated on a shaker overnight at 4°C, then centrifuged and washed with PBS-0.1% Tween20 using a magnetic plate-washer (Bio-Plex Pro wash station). The anti-human Ab detectors for IgG, IgG1, IgG2, IgG3, IgG4, IgA1, IgA2, or IgM conjugated with phycoerythrin (PE) (SouthernBiotech, Birmingham, AL) were added, and the mixture was incubated for 2 hr on a plate shaker. C1q (MP Biomedicals), FcγRI (R&D Systems), and FcγRIIIb (R&D Systems) were biotinylated using EZ-Link Sulfo-NHS-LC-Biotin (Thermo Fisher Scientific) as per manufacturer's instructions and then labeled streptavidin-PE (SouthernBiotech), before being added to samples. For FcγRIIa and FcγRIIIa, the beads/plasma samples mixture was incubated with biotinylated recombinant dimeric FcγRIIa and FcγRIIIa ectodomains (*Wines et al., 2016*) for 1 hr to allow binding of FcγRs to antigen-specific IgG, washed, and incubated with streptavidin-PE for 1 hr. After washing with PBS and resuspending in xMAP sheath fluid (Life Technologies, Scoresby, Australia), the plates were read on a Bio-Plex MAGPIX multiplex reader (Bio-Rad, Gladesville, Australia) and analyzed on Bio-Plex Manager software (Bio-Rad). A single measurement of each sample for each antibody feature was done. Data were excluded if the number of beads acquired was too low. Readouts were expressed as the raw median fluorescence index (MFI).

## Detection of antibody binding to the IE

Binding of IgG and IgG subclass antibodies to the surface of CS2 and CSA-selected 3D7 IEs was measured as previously described; all samples were run in duplicate (*Aitken et al., 2010*; *Elliott et al., 2005b*).

## ADCP of VAR2CSA DBL-coated beads by THP-1

Antibody-dependent cellular phagocytosis (ADCP) of beads coated with VAR2CSA DBL domains and opsonized with plasma or purified IgG was assessed using THP-1 cells. Individual DBL domains (*Supplementary file 1*) were biotinylated and incubated with neutravidin 1 µm beads at a ratio of 1 µL of beads (stock 1% v/v) to 3 µL of 1 mg/mL of the respective DBL domain. Beads and proteins were incubated overnight at 4°C on a roller. After incubation, beads were washed twice in PBS-0.1% BSA and then resuspended at 0.01% v/v in PBS-0.1% BSA with 0.02% sodium azide. For the phagocytosis assay, a sterile 96-well U bottom plate (Corning, Mulgrave, Australia) was blocked with high salt PBS buffer (PBS plus 0.5 M NaCl with 1% BSA 0.05% Tween20) for 30 min at room temperature. Then 20 µL of plasma diluted 1:800 in high salt PBS buffer and 10 µL of beads was added and incubated for 1 hr at room temperature. The plate was spun at $4700 \times g$ for 3 min, supernatant was replaced with 30 µL of leukocyte media (RPMI-1640 with 10% FBS and Penicillin-Streptomycin-Glutamine [all from Gibco, Scoresby, Australia]), then $5 \times 10^4$ THP-1 cells in 50 µL of leukocyte media were added and incubated for 40 min at 37°C in 5% $CO_2$. After incubation, cells were washed and

resuspended in cold PBS-4% paraformaldehyde (PFA). Phagocytosis was measured by gating on THP-1 using forward scatter (FSC) and side scatter (SSC) parameters. Phagocytic score was calculated as (% of THP-1 cells positive for beads × geometric MFI of the bead-positive THP-1 cells). All samples were run in duplicate.

## ADCP of IE by THP-1

ADCP of IE opsonized with plasma by THP-1 cells was assessed as previously described (*Ataíde et al., 2010*). Phagocytosis of IE opsonized with purified IgG was measured as previously described (*Ataíde et al., 2010*) with the modification that IE were opsonized with purified IgG at a concentration of equivalent to a 1:20 dilution of plasma. All samples were run in duplicate.

## ADCP of VAR2CSA DBL-coated beads by monocytes

Monocyte phagocytosis of beads coated with VAR2CSA DBL domains and opsonized with plasma was assessed. Beads were coated with DBL domains as described above, except that neutravidin 1 μm beads were incubated with 6 μL of 1 mg/mL of each DBL domain. To measure ADCP, a sterile 96-well U bottom plate (Corning) was blocked with salt PBS buffer (PBS plus 0.25 M NaCl and 1% BSA) for 30 min at room temperature. 40 μL of plasma (diluted 1:300 in salt PBS buffer) was added to each well along with 20 μL of beads (0.01% v/v) and incubated for 1 hr at room temperature. The beads were then washed and resuspended in 60 μL of leukocyte media. In a separate 96-well U bottom plate, $5 \times 10^4$ monocytes in leukocyte medium were plated out per well and rested at 37°C for 1 hr prior to phagocytosis. 30 μL of the opsonized beads were then added to each well with monocytes. The monocytes and beads were co-incubated for 50 min at 37°C with 5% $CO_2$. After incubation, the cells were washed and resuspended in cold PBS-2% PFA. Phagocytosis was measured by gating monocytes on the FSC and SSC. All samples were run in duplicate, and assays were repeated with three different primary cell donors. For each donor, a phagocytic score was calculated (% of monocytes cells positive for beads × geometric MFI of the bead-positive monocytes) and then the average of the phagocytic scores for the three donors was taken.

## ADCP of IE by monocytes

ADCP of opsonized IE by primary monocytes was assessed. A sterile 96-well U bottom plate (Corning) was blocked with PBS-1% BSA. In each well, 30 μL of diluted plasma (1:10) along with 3.3 μL of DHE-stained IE in leukocyte media ($1.65 \times 10^7$ IE/mL) was mixed and left for 1 hr at room temperature, before being washed and resuspended in 50 μL RPMI-1640 with 25 mM HEPES, 0.5% Albumax (w/v), 5% heat-inactivated human serum (HIHS). In a separate 96-well U bottom plate, $5 \times 10^4$ monocytes in leukocyte medium were plated out per well and rested at 37°C for 2 hr prior to phagocytosis. Then 25 μL of the opsonized IE were added to each well with monocytes. The monocytes and IE were co-incubated for 40 min at 37°C with 5% $CO_2$. After incubation, the cells were washed and resuspended in cold PBS-2% PFA. Phagocytosis was measured by gating on monocytes using the FSC and SSC parameters. All samples were run in duplicate, and assays were repeated with three different primary cell donors. For each donor, a phagocytic score was calculated (% of monocytes cells positive for DHE × MFI of the DHE-positive monocytes) and then the average of the phagocytic scores for the three donors was used.

## ADNP of VAR2CSA DBL-coated beads

Antibody-dependent neutrophil phagocytosis (ADNP) of beads coated with VAR2CSA DBL domains and opsonized with plasma was assessed. Beads were coated with DBL as described for primary monocytes. Neutravidin 1 μm beads were incubated with 6 μL of 1 mg/mL of the respective DBL domain. To quantitate ADNP, 12 μL of plasma (diluted 1:100 in high salt PBS) was added to each well of a sterile 96-well U bottom plate (Corning) along with 6 μL of beads (0.01% v/v) and incubated for 1 hr at room temperature. The beads were then washed and resuspended in 30 μL of leukocyte medium. In a separate 96-well U bottom plate, $3 \times 10^4$ neutrophils in leukocyte medium were plated out per well, 15 μL of the opsonized beads were then added to each well. The plate was incubated for 1 hr at 37°C in 5% $CO_2$, then the cells were washed and resuspended in cold PBS-2% PFA. Phagocytosis was measured by gating neutrophils on the FSC and SSC. All samples were run in duplicate, and assays were repeated with three different primary cell donors. For each donor, a

phagocytic score was calculated (% of neutrophil cells positive for beads $\times$ geometric MFI of the bead-positive neutrophils) and then the average of the phagocytic scores for the three donors was used.

## ADNP of IE

ADNP of IE opsonized with plasma was assessed. 30 µL of diluted plasma (1:10) and 3.3 µL of DHE-stained IE in neutrophil medium ($1.65 \times 10^7$ IE/mL) were mixed in a sterile 96-well U bottom plate (Corning), incubated for 1 hr at room temperature, before being washed and resuspended in 50 µL of leukocyte medium. In a separate 96-well U bottom plate, $2.5 \times 10^4$ neutrophils in 50 µL of leukocyte medium were plated out per well, 25 µL of the opsonized IE were then added to each well, and the plate was incubated for 1 hr at 37°C with 5% $CO_2$. After incubation, the cells were washed and resuspended in cold PBS-2% PFA. Phagocytosis was measured by gating neutrophils on the FSC and SSC. All samples were run in duplicate, and assays were repeated with three different primary cell donors. For each donor, a phagocytic score was calculated (% of neutrophil cells positive for beads $\times$ geometric MFI of the bead-positive neutrophils) and then the average of the phagocytic scores for the three donors was used.

## ADRB using VAR2CSA DBL domains

Antibody-dependent respiratory burst (ADRB) was measured using an assay of reactive oxygen species (ROS) production. Firstly, 25 µL of 4 µg/mL of individual DBL domains (*Supplementary file 1*) in PBS were coated on 96-well, white, flat bottom plates (NUNC MaxiSorp flat bottom; Thermo Fisher Scientific) and left overnight at 4°C. The liquid was removed, and plates were then washed with PBS and blocked with PBS-0.1% BSA for 1 hr at room temperature. Then 25 µL of each diluted sample (1:10 in PBS) was added to a single well and incubated at room temperature for 2 hr, the plates were then washed twice with PBS. Neutrophils were resuspended in Hanks' buffered saline solution at $2 \times 10^6$/mL and 20 µL of neutrophils were added to each well followed by 20 µL luminol horseradish peroxidase (HRP) solution (PBS with 33 ng/mL of HRP and 4 mM luminol; all from Sigma-Aldrich). The plate was spun briefly to settle the contents and read immediately on a FLUOstar plate reader. Luminescence was measured in each well for 1 s every 2 min for 1 hr and was calculated as the average luminescence 5 min either side of the peak of the curve. Luminescence score was standardized to the no serum controls that were run on every plate. All assays were repeated with three different primary cell donors, and the average luminescence score from the three donors was used.

## ADRB to IE

To measure ADRB in response to IEs, 96-well U bottom plates (Corning) were coated with PBS-1% FBS for 1 hr. 1 µL of plasma from each donor was aliquoted into a single well and along with 20 µL of purified IEs at $2 \times 10^7$ cells/mL in PBS-1% FBS were mixed and incubated for 1 hr at room temperature. The IE were then washed and resuspended in 20 µL of PBS. 10 µL of neutrophils in Hanks' buffered saline solution at $2 \times 10^6$/mL, 10 µL of luminol HRP solution, and 5 µL of opsonized IEs were added to each well of a 384-well white, flat bottom plate (NUNC MaxiSorp flat bottom; Thermo Fisher Scientific). The plate was spun briefly to settle the contents and then read immediately on a FLUOstar plate reader (BMG LABTECH, Mornington, Australia). Luminescence was measured as described above. Luminescence score was calculated as the % of the positive plasma control, which was run on every plate. All assays were repeated with three different primary cell donors, and the average luminescence score from the three donors was used.

## ADCC using VAR2CSA DBL domains

Antibody-dependent cellular cytotoxicity (ADCC) assays using human NK cells were modified for use with DBL antigens (*Supplementary file 1*; *Jegaskanda et al., 2013*; *Lu et al., 2016*). NUNC MaxiSorp flat bottom plates (Thermo Fisher Scientific) were coated with DBL proteins (200 ng/well) at 4°C for 12 hr. After washing with PBS, the plate was blocked with PBS-1% BSA for 1 hr. Purified IgG (0.5 mg/mL) was added to each well and incubated at 37°C for 2 hr. NK cells ($0.25 \times 10^6$ cells/mL), anti-CD107a-allophycocyanin (APC)-H7 (BD) brefeldin A (10 mg/mL; Sigma-Aldrich), and GolgiStop (BD) were added to each well, and the plates were incubated for 5 hr at 37°C. NK cells were then stained for surface markers using anti-CD16-Brilliant violet (BV)-605 (BD), anti-CD56-Brilliant

ultraviolet (BD), and anti-CD3-peridinin-chlorophyll-protein (PerCP; BD), and then stained intracellularly with anti-IFNγ-PE (BD) and anti-TNFα-BV-785 (BD) after fixation (10% PFA) and Perm B solutions (Thermo Fisher Scientific). NK cells were analyzed via flow cytometry and defined as $CD3^-$ and $CD56^{high}CD16^{\pm}$ and $CD56^{low}CD16^{high}$. Boolean gates (FlowJo) were used to include all NK cells that expressed degranulation marker CD107a or produced cytokines, IFNγ and TNFα. Levels of CD107a, IFNγ, and TNFα expression or production were calculated as % of NK cells positive for APC-H7, PE, and BV-785, respectively. All assays were repeated with three different primary cell donors, and the average score from the three donors was used.

## CSA binding inhibition antibody to IE

CSA binding inhibition was measured as previously described (*Nielsen and Salanti, 2015*). The assays were all run twice and averages of the two runs were used.

## IgG levels to non-pregnancy-specific antigens

IgG levels towards schizont extract and MSP-1 were measured by ELISA as previously described (*Barua et al., 2019*) using goat anti-human IgG biotinylated (Mabtech 3820-4-250).

## Quantification and statistical analysis

### Processing of data

Prior to analysis, the data were processed. The right skewness of the distribution of the features was reduced by log-transformation (log(x + 1)). Four antibody features that had negative values were right-shifted to have their minimum at zero prior to log-transformation. Next, the distributions of the features were centered and scaled to have zero mean and unit standard deviation.

Demographic and clinical characteristics of the pregnant women were described using the mean (SD) for continuous variables and frequency (%) for categorical variables.

## Univariate analysis

Antibody levels for individual antibody features were compared between groups using the Welch's *t*-test. The fold-change in the volcano plot was defined as the exponential of difference between the means of log-transformed and standardized data across the groups. Antibody features were correlated using Pearson pairwise correlation and correlation networks were plotted using the qgraph package (*Epskamp et al., 2012*). Associations between select antibody features and gravidity were investigated using linear regression.

## Identification of key antibody features

Non-infected women were excluded from the multivariate analyses. Multivariate imputations by chained equations (*Buuren and Groothuis-Oudshoorn, 2011*) with predictive mean matching was used to impute any missing values (0.82% of observations). The imputation process was repeated five times, and the median of the imputed values across the five generated imputed datasets was finally used for each missing value. An elastic net-regularized logistic regression (ENLR) model was initially used to identify the subset of antibody features that best discriminate between the pregnant women with non-placental infection and women with placental infection. Elastic net (*Zou and Hastie, 2005*) allows to attain sparsity (keeping only a small subset of important variables), but unlike the least absolute shrinkage and selection operator (LASSO) penalization, groups of highly correlated important variables can as well be selected (a property of ridge regression). The parameter α, which alters the nature of penalization between ridge regression and LASSO, was set to 0.5 to achieve a balance between sparsity and group selection; the final selected antibody features did not alter when $\alpha$ was varied over [0, 1], as detailed below. We used a resampling approach in which the ENLR model was repeatedly fitted to subsets of data, as used by *Gunn et al., 2018*. This provided a means to take account of the uncertainty in the feature selection by ENLR and identify the features that are consistently selected when the model is fitted to different resampled data. The selection stability of an antibody feature was defined as the proportion of times that it was picked in the selected set of important features by elastic net when the model was repeatedly fitted to 50,000 resampled subsets of data (5000 repeats of 10-fold cross-validation). In each of the 5000 repetitions, the dataset was divided into 10-folds, of which 9-folds were used to perform an inner 10-fold cross-

validation to find the best value of λ (shrinkage parameter) that maximizes the area under the receiver operating characteristic (AUROC) and in turn select the antibody features; using other performance measures such as binomial deviance gave the same ranking of the features according to selection frequencies. In each resampling iteration, the majority group (50 women with placental malaria) was downsampled to have an equal number of observations in both groups, thus avoiding overfitting the model to the majority group. The resampling process was repeated where $\alpha$ was also tuned over the set {0, 0.25, 0.5, 0.75, 1} (instead of only $\alpha = 0.5$) in addition to λ, and the results showed that the final top frequently selected variables do not change (see *Figure 2—figure supplement 1*).

Partial least squares discriminant analysis (PLSDA) (*Barker and Rayens, 2003*) was subsequently applied on the results of the ENLR to reassert the set of selected antibody features and find a minimal set of antibody features that accurately classifies the pregnant women; two components were selected for the PLSDA model (exception: one component was used where the model included only one feature). The antibody features were added to the PLSDA model one by one, from the highest to lowest selection frequency as estimated by the ENLR, and the minimal set of features beyond which the accuracy did not improve significantly were selected as the final set of key antibody features (*Gunn et al., 2018*); 500 repeats of 10-fold cross-validation were performed to compute the accuracies. The statistical significance of the results was assessed by comparing the performance of the model with that of two random permutation tests (null cases): (1) the PLSDA model was fitted to six randomly selected antibody features and the performance was computed for 500 repeats of 10-fold cross-validation resampling; (2) 100 datasets were generated by randomly permuting the group labels (PM and NPI) and the same analysis performed for the original dataset (i.e., building PLSDA models using the top six frequently selected antibody features found by resampling of elastic net) was repeated for each dataset.

All of the computations were performed in the R software (*R Development Core Team, 2019*). The resampling was performed using the caret package (*Kuhn, 2008*). The glmnet package (*Zou and Hastie, 2005*) was utilized for implementing the ENLR. The PLSDA method was implemented via the PLS package (*Mevik and Wehrens, 2007*), and some visualizations of the results of the PLSDA method were carried out using the mixOmics package (*Rohart et al., 2017*).

# Results

## Cohort description

We measured antibody features in plasma samples collected at enrollment (mean 21 weeks' gestation) in 127 women and assessed the influence of antibody features on the presence of placental malaria at delivery. All women resided in malaria-exposed areas, and 17 were positive for *P. falciparum* at enrollment (by PCR and/or light microscopy). Women were selected based on *P. falciparum* infection status at delivery, using histology to define placental malaria (*Bulmer et al., 1993*). The cohort comprised 50 women who subsequently had no *P. falciparum* parasites detected on placental histology and were PCR negative for peripheral parasitemia at delivery (non-infected), 50 women who had *P. falciparum* parasites in the placenta by histology (placental malaria), and 27 women who had no *P. falciparum* parasites in the placenta by histology but who had peripheral blood parasitemia by PCR and/or light microscopy (non-placental infection) at delivery. Of the latter group, 10 were also PCR positive in placental blood, consistent with free circulation of early-stage parasites throughout the body. SP and azithromycin was more efficacious against placental malaria than SP-chloroquine (*Unger et al., 2015*), but antimalarial regimes were similarly distributed between groups. The three groups were frequency matched for primigravidity, IPTp regime receipt, bed net use, rural residency, and age, and all three groups had similar clinical characteristics at enrollment (*Table 1*).

## Measurement of antibody features

We initially studied antibody to an array of recombinant VAR2CSA DBL proteins (see *Figure 1—figure supplement 1* for a schematic of VAR2CSA). Thirteen features of antibody towards recombinant forms of each of the six DBL domains were measured by multiplex in the 127 women. In order to consider antigenic diversity (*Benavente et al., 2018*), four of the DBLs were also represented by an

additional recombinant protein from a heterologous parasite strain (see *Supplementary file 1* for details of recombinant proteins). Antibody features measured by multiplex included antibody isotypes (IgG, IgA, IgM) and subclasses (IgG1, IgG2, IgG3, IgG4, IgA1, IgA2), and the ability of antigen-specific antibody to bind Fcγ receptors (FcγR) and complement protein C1q (which assesses the ability of antibodies to activate the classical complement cascade). Thus, using samples from the 127 women and measuring 13 antibody features directed at 10 DBL domains, 15,340 out of 16,510 possible data points were acquired (see *Figure 1—figure supplement 2* for a flow chart of data acquisition).

Levels of antibody features measured by multiplex were compared between non-infected and placental malaria, and between non-placental infection and placental malaria groups. Volcano plots summarizing the two comparisons of the multiplex data are given in *Figure 1* (see *Supplementary file 2* for individual univariate analyses). Notably, levels of antibody to DBL domains at enrollment were similar between non-infected women at delivery and those with placental malaria (*Figure 1A*), whereas multiple antibody features were higher in women with non-placental infection than in placental malaria, indicating potential roles in protection (*Figure 1B*). Antibody features that were higher in women with non-placental infection compared to those with placental malaria included DBL2-specific IgG3, IgA2, IgG2, and IgG4 (IgG3.DBL2(ID1-ID2a).FCR3, IgA2.DBL2.1010, IgG2.DBL2(ID1-ID2a).FCR3, and IgG4.DBL2.1010, respectively) as well as DBL3-specific antibodies that bind FcγRIIIb (FcγRIIIB.DBL3.FCR3) and DBL5-specific IgG2 and IgG4 (IgG2.DBL5.0466 and IgG4.DBL5.0466). Thus, we selected DBL2, DBL3, and DBL5 for the assessment of functional antibody features using cell-based assays in order to provide insights into the mechanistic contribution of these enhanced antibody populations. To investigate whether the higher antibody levels seen in the non-placental infection group were just correlates of exposure, we also measured IgG levels to non-pregnancy-specific *P. falciparum* antigens merozoite surface protein-1 (MSP-1) and schizont extract (which are associated with exposure *Barua et al., 2019*) by ELISA and compared the levels

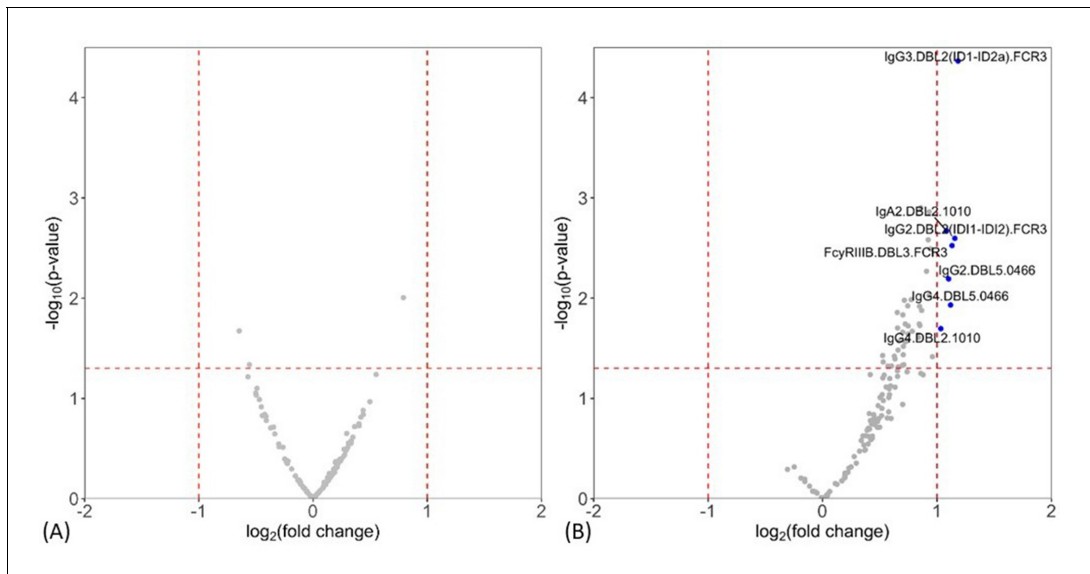

**Figure 1.** Individual antibody features to recombinant VAR2CSA Duffy binding-like (DBL) domain proteins measured by multiplex comparing women with placental malaria at delivery to (**A**) non-infected women and (**B**) women with non-placental infection. Fold-change (log$_2$ transformed), characterizing the magnitude of difference between the antibody levels of two groups (x-axis), is plotted against the -log$_{10}$ p-value, characterizing the statistical significance of the difference (y-axis). The vertical dotted lines (log$_2$(2) and log$_2$(0.5)) mark a threshold for a twofold change, and the horizontal dotted lines (log$_{10}$(0.05)) mark the statistical significance threshold (p≤0.05, Welch's *t*-test). Antibody features beyond the thresholds are shown in blue and labeled.

The online version of this article includes the following figure supplement(s) for figure 1:

**Figure supplement 1.** Diagram of the *Plasmodium falciparum* erythrocyte membrane protein 1 (PfEMP1) VAR2CSA on the surface of the infected erythrocyte.

**Figure supplement 2.** Flow chart of data acquisition and analysis.

between groups. IgG levels to neither MSP-1 nor schizont extract were higher in women without placental malaria (*Supplementary file 2*), suggesting that differences in antibody to pregnancy specific antigens were not simply due to exposure. As antibody responses did not significantly differ between non-infected and placental malaria groups, further data analysis was restricted to those 77 women with non-placental infection or with placental malaria.

We next measured responses to native protein on the surface of IEs of *P. falciparum* lines CS2 (which is isogenic with FCR3) and 3D7 that had been selected for high levels of CSA adhesion and studied whether antibody-opsonized IE and beads coated with recombinant DBL domains could activate leukocytes or promote phagocytosis to identify which antibody features associated with innate immune cells play important roles in parasite clearance or killing. We adapted existing cellular assays to measure ADCP of DBL protein-coated fluorescent beads and IE using THP-1 cells (*Ackerman et al., 2011*; *Teo et al., 2015*) and to measure NK cell activation following IgG binding to DBL proteins in plate-based assays (*Jegaskanda et al., 2014*; *Lu et al., 2016*). We also developed novel assays of ADCP by primary monocytes, and ADNP using primary human neutrophils, with IE and DBL-coated beads as their targets, as well as assays to measure ADRB with IEs or DBL proteins coated onto plates as their targets. Thus up to nine different cellular responses towards DBL2, DBL3, and DBL5 were measured. Additionally, six features of antibody to VSA of two strains of CSA-binding IE were measured, namely IgM, IgG, and IgG subclass binding to VSA, and the ability of antibodies to block CSA binding was assessed (see *Supplementary file 3* for a table listing all antibody features measured). With 39 variables for 77 women, 2997 measurements (99.8% of possible 3003) were acquired. When added to the multiplex data, 169 variables with a total of 12,906 measurements (99.2% of 13,013) were used in subsequent analyses (see *Figure 1—figure supplement 1* for the flow chart). None of the women had any data missing for the outcome variable, and 11 out of 77 women had data missing for one or more of the 169 predictor variables.

## Identifying antibody features at enrollment that differentiate women with non-placental infection and placental malaria at delivery

We used a combination of ENLR and PLSDA to identify a set of antibody features that could best differentiate women with non-placental infection from those who had placental malaria at delivery (*Gunn et al., 2018*). Initially, the ENLR model was repeatedly fitted to randomly resampled subsets of the data and the selection frequency and the effect size (odds ratio) of the antibody features were calculated (*Figure 2*); only the top 20 antibody features with highest selection frequency are shown. The results show the antibody features that were consistently selected (*Figure 2A*) and have a high effect on discriminating the group of pregnant women (*Figure 2B*). The frequently selected antibody features were positively associated with non-placental infection (*Figure 2B*), and the effect of the antibody features on the odds of belonging to the non-placental infection group increased as the selection frequency increased.

A minimal subset of antibody features that could accurately separate the two groups was then identified by adding the antibody features into a PLSDA model one by one (based on the selection frequency; *Figure 2A*). The mean AUROC and accuracy of classifying women with non-placental infection do not increase significantly after the inclusion of the six top features (*Figure 3A*). The performance of the model with these six selected features was then compared with the results of two random permutation tests (*Figure 3B*). The results show that the model with the six selected features performed significantly better than the PLSDA models with randomly selected features. The AUROC is also significantly higher than that of the permutation test where the group labels were randomly permuted (*Figure 3B*). The AUROC for the six features corresponds to a median accuracy of 0.86 (*Figure 3—figure supplement 1*) and means that the model with these six antibody features can accurately classify, on average, 86% of the women.

The segregation of the pregnant women and the loadings on the components of the PLSDA model for the six selected features are illustrated in *Figure 3C, D*. The loading components illustrate that the segregation of the observations by the PLSDA was largely driven by the top two variables identified by ENLR, DBL2-specific IgG3 (IgG3.DBL2(ID1-ID2a).FCR3), and ADCP of the CSA-binding IE CS2 by THP1 (THP1.Phago.CS2).

As immunity against placental malaria increases with gravidity (*Fried et al., 1998*), we also conducted a univariate linear regression for these six selected variables with gravidity (*Supplementary file 4*). Ability of antibodies to inhibit binding of IE to CSA (CSA.Binding.Inhibition.

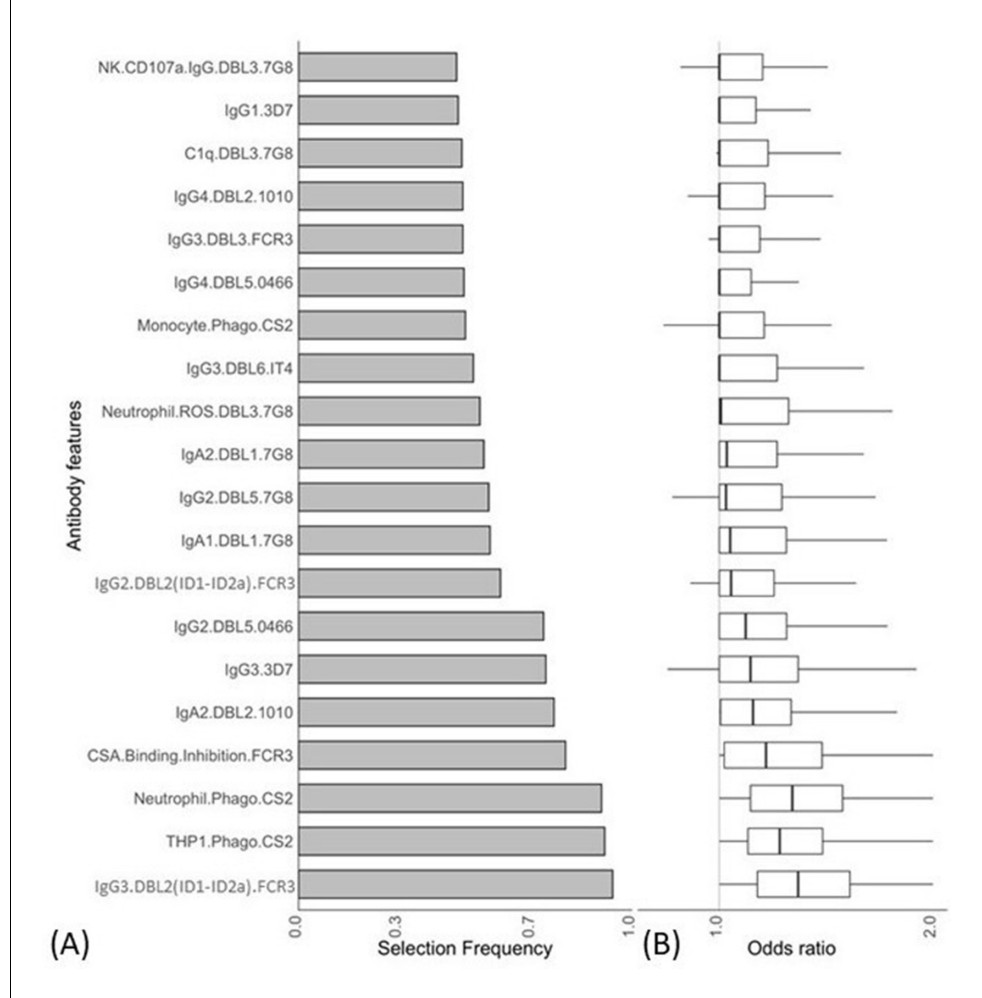

**Figure 2.** Antibody features that are influential in distinguishing between malaria-infected women with and without placental infection identified by the elastic net-regularized logistic regression model. Resampling (5000 repeats of 10-fold cross-validation) was used to obtain the selection frequencies and the odds ratios. (**A**) The top 20 antibody features are ranked in ascending order of selection frequency. (**B**) Boxplots of the estimated odds ratios, an odds ratio >1 indicates the antibody feature is positively associated with non-placental infection at delivery. Boxplots are median, IQR, whiskers (the lowest data point that falls between Q1 and 1.5 * Q1 IQR, the highest data point that falls between Q3 and 1.5 * Q3 IQR). IQR: interquartile range.

The online version of this article includes the following figure supplement(s) for figure 2:

**Figure supplement 1.** Selection frequencies estimated across all $\alpha$ values {0, 0.25, 0.5, 0.75 1} using the resampling of elastic net-regularized logistic regression.

FCR3) was the only antibody feature with evidence of a positive association with gravidity in this cohort (p=0.046). In addition, as a small subset of women were infected with *P. falciparum* at enrollment (by light microscopy and/or PCR, n = 17, **Table 1**), to investigate the influence of these early infections on the identified features we conducted a subanalysis, excluding these infected women, comparing the six identified antibody features between non-placental and placental malaria cohorts (**Supplementary file 4**). Importantly, four of the six antibody features were significantly higher in the non-placental infection cohort (p≤0.013), and the remaining two features trended to be higher in the non-placental cohort (p≤0.064). This suggested that the associations seen with these six variables were not greatly affected by the presence of the individuals with infections at enrollment.

Closer inspection of the features using a correlation network of the 77 women provided information about the relationships between different antibody features (**Figure 4**). Distinct clustering of some antibody features was apparent; for example, IgA1, IgA2, IgG4, and IgM to recombinant DBL

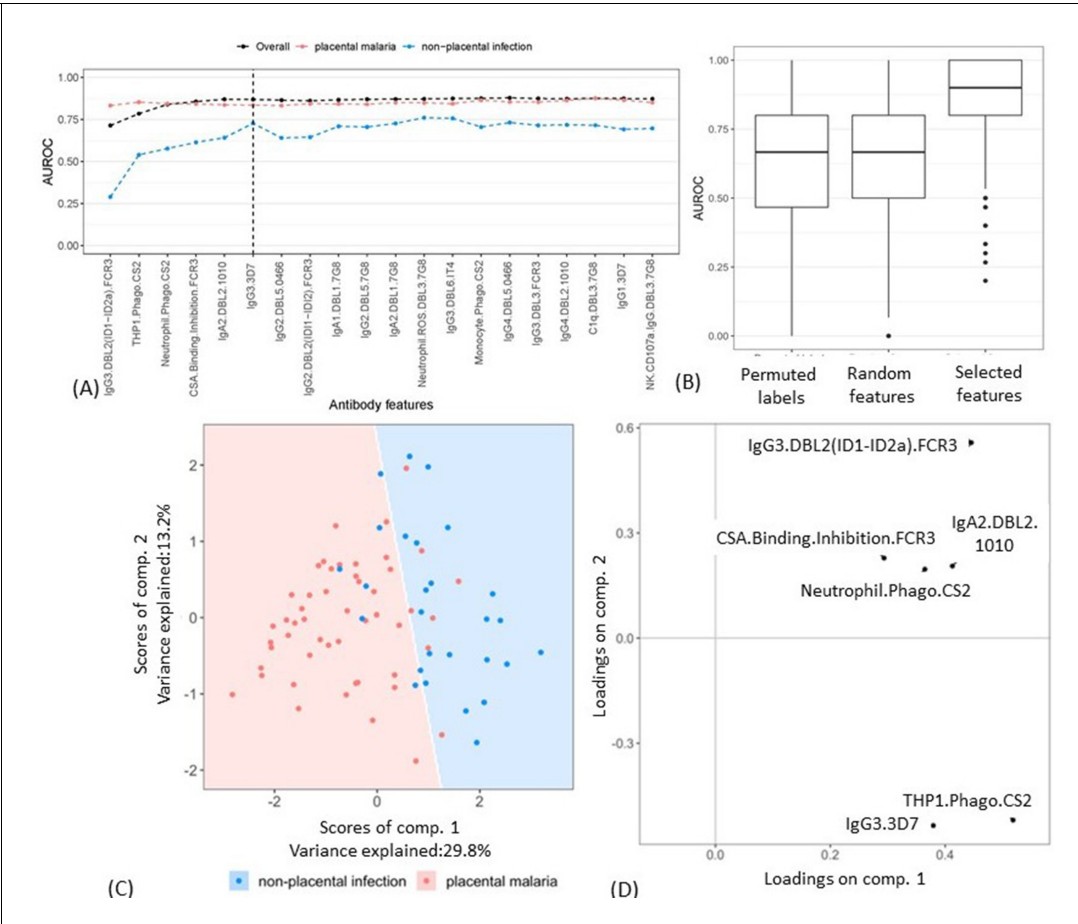

**Figure 3.** Selecting a minimal set of antibody features by partial least squares discriminant analysis (PLSDA). (**A**) Performance of PLSDA at classifying women as having placental malaria (PM) or non-placental infection (NPI) when the features (ranked by selection frequency using the elastic net; *Figure 2*) are added one by one to the model from highest to lowest rank (500 repeats of 10-fold cross-validation were performed to estimate accuracy for each model). The three lines represent the accuracy of classification of all women in the cohort (black), those with NPI (blue), and those with PM (red). The vertical dashed line denotes the cutoff beyond which the accuracy does not change significantly by adding more antibody features (used for selecting the minimal set of features). (**B**) Comparing the performance of the elastic net-regularized logistic regression (ENLR) + PLSDA applied on the original data of the top six variables (rightmost boxplot) with two permutation tests (null cases): (1) the PLSDA model was fitted to six randomly selected antibody features and the performance was computed for 500 repeats of 10-fold cross-validation resampling; (2) 100 datasets were generated by randomly permuting the group labels (PM and NPI) and the same analysis performed for the original dataset (i.e., building PLSDA models using the top six frequently selected antibody features found by resampling of elastic net) was repeated for each dataset. (**C**) Segregation of women with NPI (blue) and PM (red) using the scores of the two components of the PLSDA model with data from the selected six antibody features. The background colors show the predicted classification of the women for all the possible score values in the depicted range. (**D**) Feature loadings on the components of the PLSDA of the six selected antibody features (see *Figure 3—source data 1* for more details about the factor loadings and group prediction using the PLSDA). Boxplots are median, IQR, whiskers (the lowest data point that falls between Q1 and 1.5 * Q1 IQR, the highest data point that falls between Q3 and 1.5 * Q3 IQR). IQR: interquartile range; AUROC: area under the receiver operating characteristic curve.

The online version of this article includes the following source data and figure supplement(s) for figure 3:

**Source data 1.** Partial least squares discriminant analysis (PLSDA) prediction model.

**Figure supplement 1.** Accuracy of partial least squares discriminant analysis (PLSDA) at classifying women as having placental malaria or non-placental infection when the features (ranked by selection frequency using the elastic net; *Figure 2*) are added one by one to the model from highest to lowest rank (500 repeats of 10-fold cross-validation were performed to estimate accuracy for each model).

domains formed four distinct clusters. Similar functions sometimes also clustered together (e.g., primary monocyte and THP1 cell phagocytosis of DBL domain-coated beads). Multiplex measurement of IgG, IgG1, IgG3, and ability of antibody to bind FcγRs and C1q for all the different DBLs formed one large cluster, highlighting the collinearity of many of the antibody features measured towards recombinant proteins. Importantly, the six features identified by ENLR + PLSDA were spread

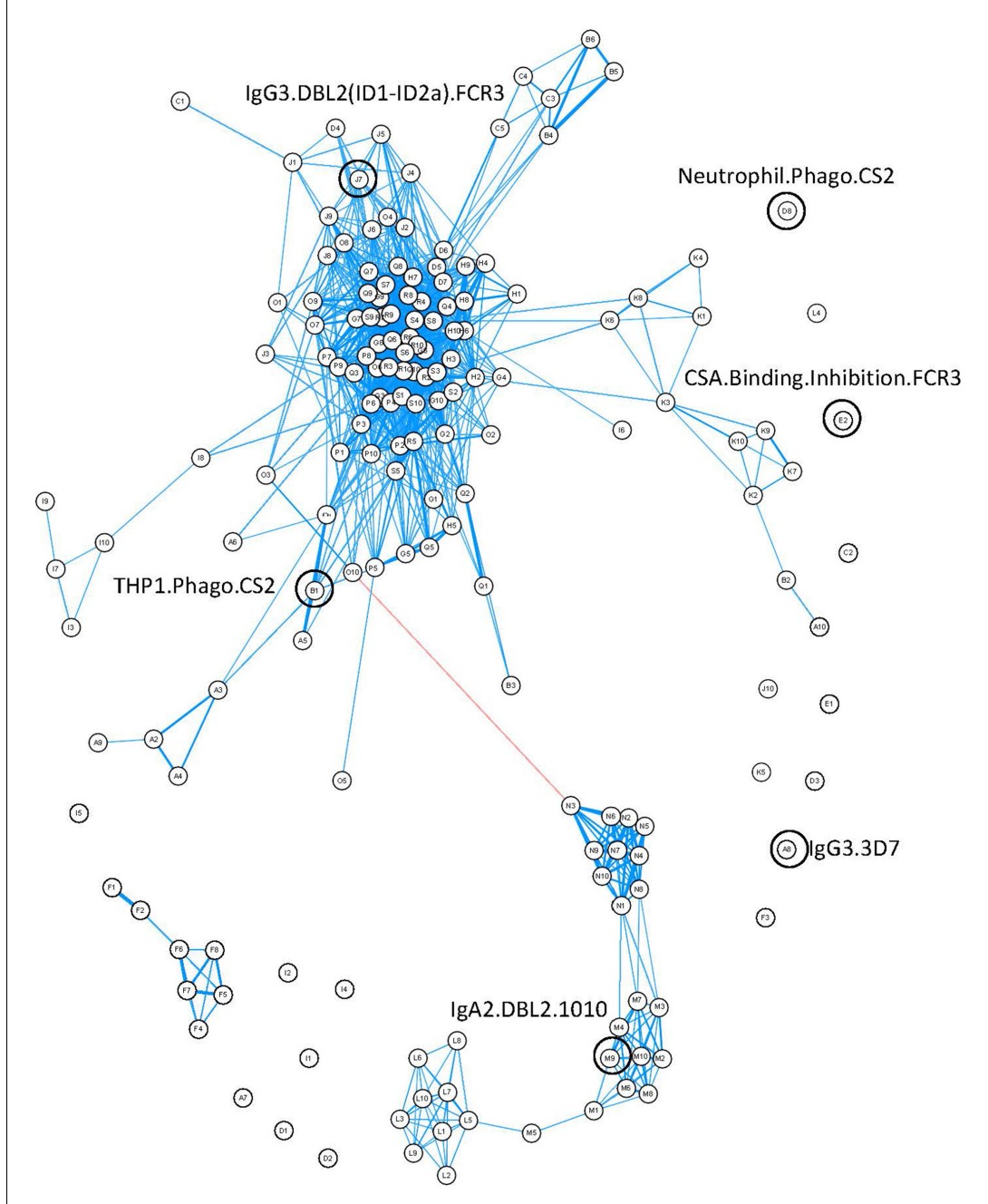

**Figure 4.** Correlation network of antibody features. Correlation network of all antibody features in both women with non-placental infection and those with placental malaria. The six selected antibody features do not cluster together. Antibody features with similar functions (denoted by the same letter) tend to correlate with each other. Blue: positive correlation; red: negative correlation; line width and closeness of variables increase with increasing correlation coefficients; only significant correlations (after Bonferroni correction for multiple comparisons) are shown. Selected antibody features identified by elastic net are highlighted and labeled. See *supplementary file 3* for a full list of feature names.

The online version of this article includes the following figure supplement(s) for figure 4:

**Figure supplement 1.** Correlation matrix of the six selected antibody features from women with non-placental infection at delivery and women with placental malaria.

throughout the correlation network and were not well correlated with each other (*Figure 4*, *Figure 4—figure supplement 1*).

## Antibody features selected highlight the importance of functional antibodies towards the infected erythrocyte and DBL2

The six selected features included IgG3 to DBL2 (IgG3.DBL2(ID1-ID2a).FCR3), ADCP, and ADNP of CSA-binding IE (THP1.Phago.CS2 and Neutrophil.Phago.CS2, respectively), binding inhibition of IE to CSA (CSA.Binding.Inhibition.FCR3), IgA2 to DBL2 (IgA2.DBL2.1010), and IgG3 to CSA binding IE (IgG3.3D7). For all six features, antibody levels were significantly higher in the non-placental infection group compared to the placental malaria group (*Figure 5A–F*), but the patterns of antibody responses distinguishing the two groups were not uniform. For example, some women in the non-placental infection group had no measurable levels of THP1.phago.CS2, IgG3.3D7 or IgG2.DBL5.0466 antibody. This indicated that there are multiple mechanisms for antibodies to confer protection. To investigate this further, we constructed a heat map of the Z-score of each feature against individual women and compared the groups with non-placental infection and placental malaria (*Figure 5G*). Within these two cohorts, there was significant variation in individual profiles and no evidence for a unique and consistent profile of antibody functions in all individuals with non-placental infection. Overall, the women with non-placental infection had significantly higher levels of most of the identified antibody features; 81% of women with non-placental infection had a Z-score >0 for at least four antibody features (compared to 26% of those with placental malaria). Looking at just the three functional measures, 52% of women with non-placental infection had elevated antibodies (a Z-score >0) that both promoted IE phagocytosis by THP-1 cells and/or neutrophils and inhibited binding of IE to CSA, compared to 18% of women with placental malaria (*Figure 5G*).

## Discussion

We used a systems serology approach to broadly characterize antibody responses to the VAR2CSA protein that mediates protection from placental malaria. This involved measuring a wide range of antibody features to this protein followed by employing the machine learning techniques elastic net-regularized regression and PLSDA to identify those antibody features that protect pregnant women with *P. falciparum* infection from placental malaria. The combination of a prospective study design and a systems serology approach provided an unbiased way to identify the key features of the antibody response to VAR2CSA that mediates protection. Using this approach, we identified six antibody features of 169 tested, which were able to correctly differentiate, on average, 86% of the pregnant women with placental malaria and non-placental infection. The antibody features identified fell into broad functional groups: antibody that blocks placental binding of IEs and opsonizing antibody that result in phagocytic clearance of IEs (see *Figure 6* for a schematic of mechanisms of protection of the six selected antibody features).

We hypothesized that pregnant women who had *P. falciparum* peripheral blood parasitemia at delivery but who did not have placental sequestration were protected from placental malaria by antibody immunity targeting VAR2CSA, and we sought to determine the characteristics of this protective antibody response. We frequency matched groups on participant characteristics such as gravidity and urban or rural residence to minimize confounding by these variables that are known to affect exposure and immunity. Antibody features at enrollment did not differ between women with placental malaria and those who were uninfected at delivery, but they did differ between infected women with and without placental malaria at delivery.

Unlike previously published studies (*Aitken et al., 2010*; *Fried et al., 2018*), the inclusion of an infected group without placental malaria allowed us to accurately control for exposure to the parasite at the time point of outcome. The lack of differences in antibody features between the non-infected and placental malaria groups was quite striking and mirrors previous observations (reviewed by *Cutts et al., 2020*). It may in part be attributable to the heterogeneous nature of the non-infected group, which probably includes both women who are exposed but protected by antibody and women who have not been exposed to malaria in pregnancy and therefore lack pregnancy-specific antibodies. It is possible that the non-infected group is protected by antibodies towards other antigens. That differences were seen only when comparing the two infected groups, clearly highlighting the importance for controlling for exposure when studying antibody responses and

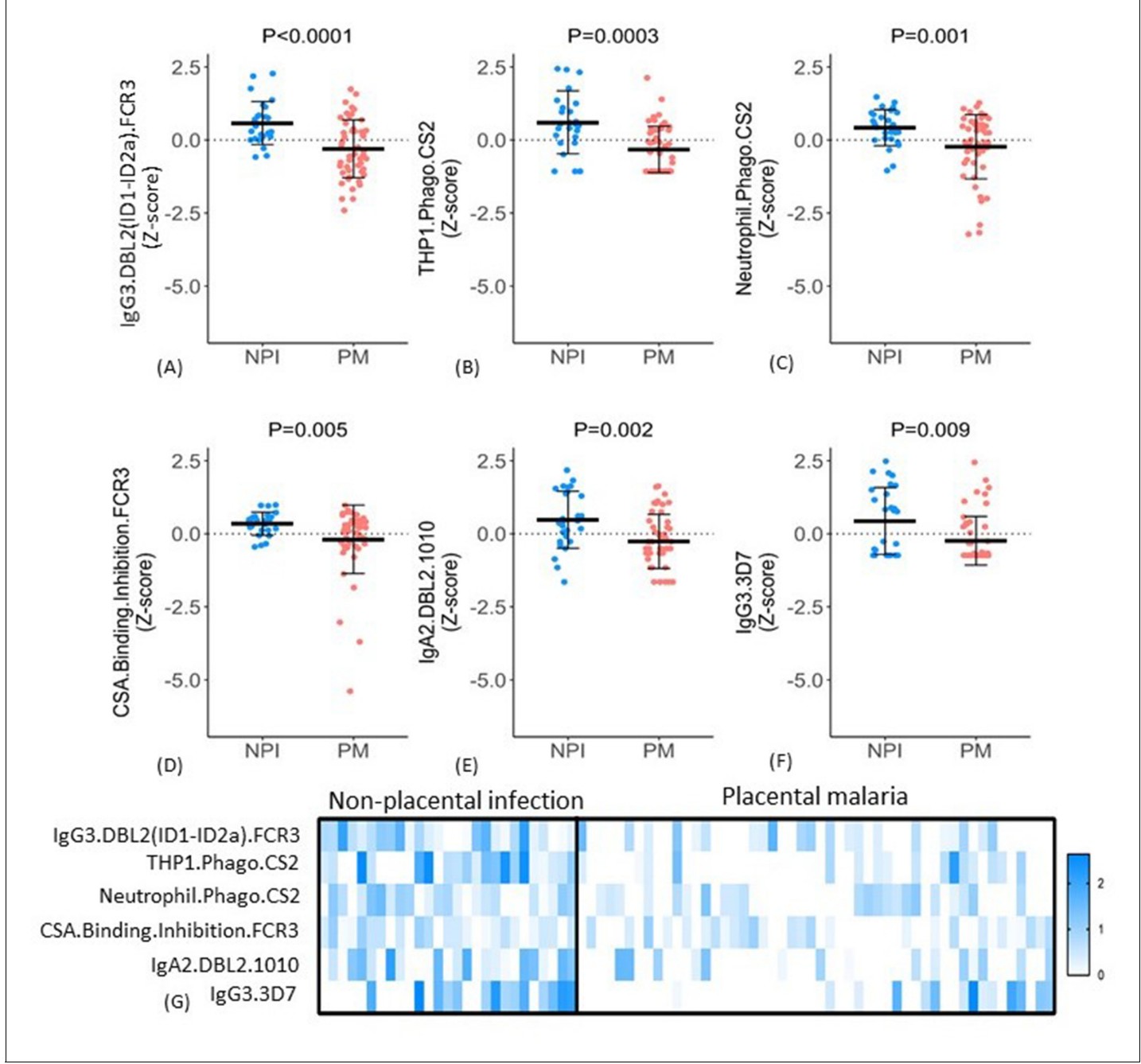

**Figure 5.** Distribution of antibody features in women with non-placental infection (NPI) and placental malaria (PM). (A–F) Levels of each of the selected antibody features in individual women in the two groups (G) No single antibody feature was present in all individuals with NPI (or was absent in all those with PM). Errors bars are mean (SD), p-values derived from Welch's *t*-test. IE: infected erythrocyte; Z-score: distribution of the features was centered and scaled to have zero mean and a standard deviation of 1.

protection from malaria. In Cameroon, multigravid women studied in early pregnancy who did not go on to have placental malaria had a greater breadth and intensity of antibody responses to DBL domains than those with placental malaria at delivery (*Tutterrow et al., 2012*), and higher avidity antibodies to full-length VAR2CSA, suggesting that a broad, high-affinity IgG response to DBL proteins may also correlate with protection.

In this study, fitting the ENLR model to resampled subsets of data allowed us to rank antibody features by their selection frequency. We generated a ranking of the 20 most frequently selected

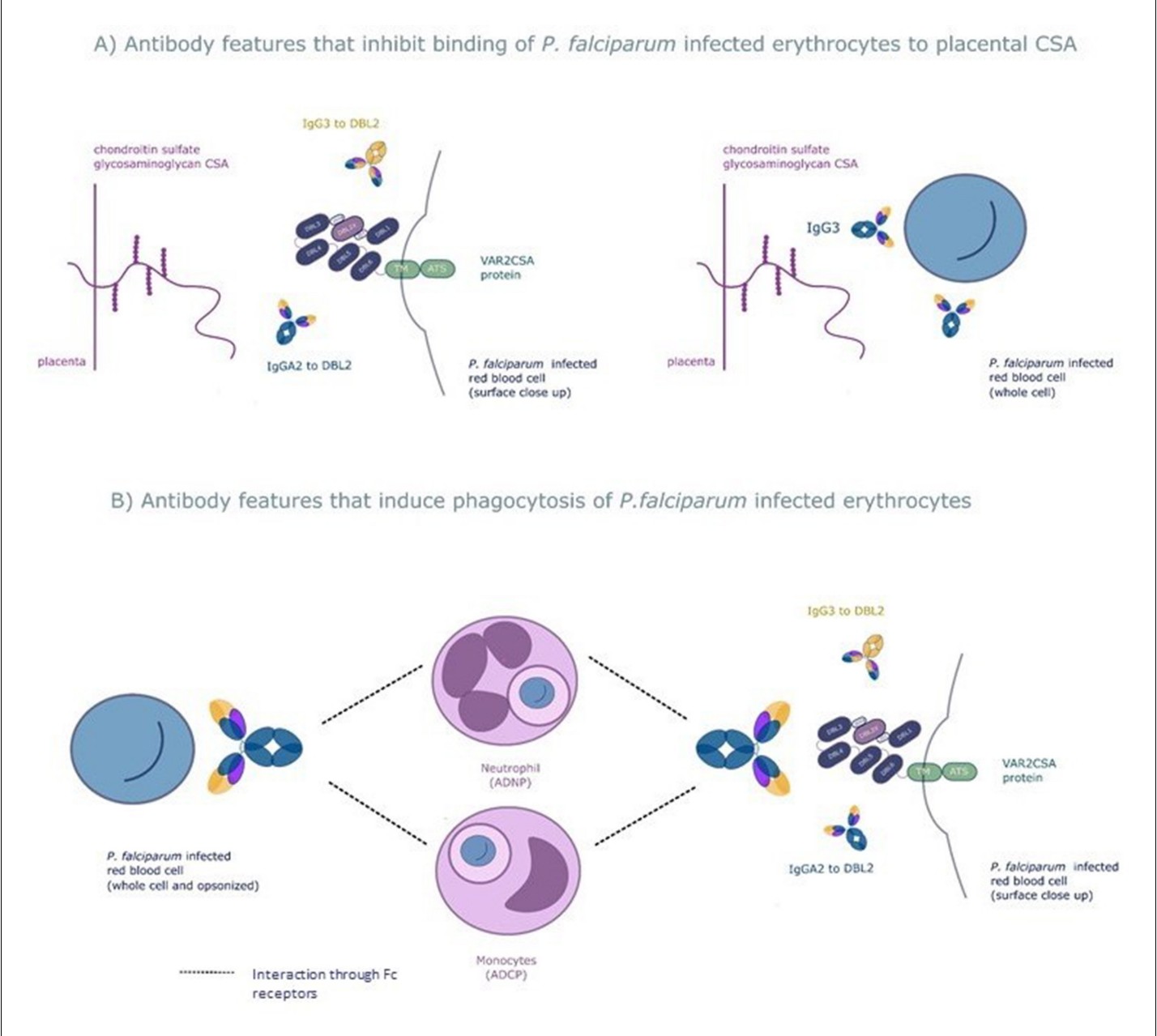

**Figure 6.** The six selected antibody features may protect women from placental malaria by (**A**) inhibiting infected red blood cells from binding chondroitin sulfate A (CSA) and sequestering in the placenta and/or (**B**) promoting phagocytosis of infected red blood cells by monocytes and/or neutrophils. Selected features that may inhibit placental sequestration include IgG3 to the whole infected red blood cell and IgG3 and IgA2 to VAR2CSA's CSA binding domain DBL2. Selected features that may promote parasite clearance by antibody dependent phagocytosis include IgG3 to the whole infected red blood cell and to DBL2, IgA2 to DBL2, and neutrophil and monocyte phagocytosis of whole infected red blood cells. ADNP: antibody-dependent neutrophil phagocytosis; ADCP: antibody-dependent cellular phagocytosis; VAR2CSA: a parasite protein expressed on the surface of the infected red blood cell, made up of Duffy binding-like domains (DBL).

features, and then used PLSDA to determine a minimal set of antibody features that classify the pregnant women with a high accuracy. These six features were selected by ENLR more than 70% of the time, and the inclusion of these six features in the PLSDA showed an ability to accurately distinguish 86% of the women in the two groups. Notably this ability to distinguish the two groups did not increase if more than six features were included in the model. The six antibody features that

best distinguished the groups were all more common in women with non-placental infection, and they included previously identified and novel responses.

Among the features we identified as associated with protection against placental malaria were adhesion inhibition and cytophilic antibodies that mediate ADCP of IE. The ability to block IE binding to CSA has been associated with protection against placental malaria, maternal anemia, low birth weight, or prematurity in subsets of pregnant women (*Duffy and Fried, 2003*; *Ndam et al., 2015*). IgG1 and IgG3 antibody responses to VAR2CSA are dominant and mediate opsonic clearance of IE (*Tebo et al., 2002*; *Elliott et al., 2005a*; *Megnekou et al., 2005*; *Keen et al., 2007*; *Damelang et al., 2019*). ADCP using THP-1 cells has been correlated with protection from malaria or its complications in pregnant women and in adults (reviewed in *Teo et al., 2016*). ADNP of IE, which has not previously been systematically studied in malaria immunity, also correlated with placental infection status. Antibodies that block adhesion and mediate ADCP by THP-1 cells or ADNP are important correlates of protection that should be included in future studies of naturally acquired and vaccine-induced antibody to malaria in pregnancy.

Although only 11% of features measured antibody to IE, four of the six top antibody features involved assays using IE. Similarly, both features towards recombinant proteins involved the DBL2 domain, which is the principal DBL domain involved in placental adhesion. The overrepresentation of both IE and DBL2 in the selected features supports the notion that antibody features directed towards functionally important targets are key to protection from placental malaria.

Novel features identified using the system serology approach include the antibody feature IgA2. Of note, IgA2 levels in serum are generally low (most serum IgA is IgA1) (*Macpherson et al., 2008*). Interestingly a recent study looking for correlates of malaria protection from vaccine-induced immunity also identified IgA2 as a variable that helped predict those protected from *P. falciparum* infection (*Suscovich et al., 2020*). IgA2 can elicit myeloid cell effector functions, including opsonic phagocytosis (mediated by FcαRI) (*Breedveld and van Egmond, 2019*), and as it recognized the CSA binding DBL domain DBL2, it may also inhibit IE binding to CSA. However, since IgA2 levels in serum are low, we hypothesize that rather than being directly involved in protection IgA2 might instead be a marker of a protective antibody response. Further investigation of VAR2CSA-specific B cells and whether a protective response is associated with increased IgA2 is warranted.

Our analysis suggests that features determined by measuring antibody to VAR2CSA-expressing IE are important indicators of protection from placental malaria. Four features measured antibody to IE using representatives of the dominant 3D7-like and FCR3-like clades (*Patel et al., 2017*; *Benavente et al., 2018*) whose N-terminal regions are used in two VAR2CSA-based pregnancy malaria vaccines (*Sirima et al., 2020*; *Mordmüller et al., 2019*). While three of these four assays targeted FCR3 or the isogenic CS2 parasite line, studies of global diversity in VAR2CSA to date have included few sequences from Papua New Guinea (*Benavente et al., 2018*), and the relative importance of immunity to specific clades may vary by region. Vaccination offered limited heterologous protection (*Sirima et al., 2020*), and it is unclear whether effective naturally acquired protective immunity needs to target multiple clades or different clades in different regions.

Antibody responses to VAR2CSA domains serve as excellent markers of exposure to malaria in pregnant women (*Cutts et al., 2020*), but few were selected as correlates of protection in our analysis. This may reflect the fact that the responses to different VARCSA domains were generally highly correlated, as revealed by their clustering together in the center of our network analysis, or it might be because of the longevity of antibody that promotes ADCP of IE compared to that of antibody responses to recombinant proteins (*Teo et al., 2014*; *Teo et al., 2015*; *Chandrasiri et al., 2014*). Antibody to VAR2CSA recombinant proteins, and particularly to specific VAR2CSA peptides, appears to be a good correlate of exposure and transmission intensity (*Fonseca et al., 2019*), but these responses appear to be less useful as correlates of protection.

The correlation network showed close correlations between different antibody responses to recombinant DBL domains (which are predominantly located in the central cluster) but the protective antibody features were dispersed, suggesting that distinct pathways contribute to the development of protection against placental malaria. The analysis of functions identified in the selected features further supports this, with most protected women (and few unprotected women) having significant antibody that promoted both binding inhibition to CSA and phagocytosis; however, no consistent pattern of antibody functions in protected women was seen. Similar associations have been seen in the relationships between antibody features and protection from viral pathogens, and there is a

growing awareness that polyfunctional antibody responses, rather than a single functional correlate, contribute to protection from disease (*Gunn et al., 2018*; *Chung et al., 2014*). None of the antibody features targeted more than one variant of the protein or parasite strain. One possible explanation is that epitopes recognized by antibodies mediating each function may vary between strains (as suggested by *Doritchamou et al., 2019*). This could in turn mean that protective functional features could vary with different populations (as there is variation in global distribution of the different VAR2CSA domain clades; *Benavente et al., 2018*). Future studies should ideally include multiple VAR2CSA clades and study diverse populations.

We used systems serology to identify protective features of the naturally acquired antibody response to VAR2CSA, a PfEMP1 variant that is specifically associated with placental malaria, complementing a recent study that used systems serology to identify functional antibody correlates of vaccine-induced immunity, in subjects vaccinated with RTS,S (*Suscovich et al., 2020*). A systems serology approach could also be used to identify the characteristics of protective immunity to other malaria syndromes, such as cerebral malaria, which is often associated with a subset of PfEMP1s that are potential vaccine candidates (reviewed in *Jensen et al., 2020*). It will be important to determine whether protective features of the cerebral malaria-associated PfEMP1 antibody response analogous to the ones that we identified in this study can be identified.

We were interested in measuring antibodies to placental binding parasites and defined placental malaria by the detection of IE in the placenta by histology because previous studies have shown that these are the infections associated with adverse pregnancy outcomes (*Menendez et al., 2000*; *Rogerson et al., 2003*). Ring stage parasites in the peripheral blood will circulate through the placenta, and, in some cases, these could be detected by PCR of placental blood; however, we did not consider these submicroscopic infections as placental malaria as they were not shown to be sequestering there.

Study strengths include the application of 169 novel and established assays; minimal missing predictor and no missing outcome data, the ability to control for exposure with the identification of pregnant women with *P. falciparum* infection with, and without, placental malaria; the prospective study design; and the use of native antigens on IEs to complement the data we acquired using recombinant protein. Together, this allowed us to discover the determinants of naturally acquired protective immunity. Weaknesses include the lack of inclusion of full-length VAR2CSA in our Luminex and bead-based assays, and an imbalance in numbers between women with parasitemia but no placental infection relative to the other groups. Samples from all women in the former group were selected for this study (and statistical analysis controlled for this imbalance between groups), but this group is relatively rare, with placental infection in the absence of peripheral infection being substantially more common (*Desai et al., 2007*). High levels of collinearity between some variables make it difficult to dissect out individual protective responses. Demonstration that our selected variables correlate with lack of placental malaria in other cohorts would further strengthen the findings. Antibody glycosylation (*Lu et al., 2016*) is an additional feature that could be included in future studies.

The present study identified leading candidate antibody features that mediate naturally acquired protection from placental malaria and demonstrated we could use these features to identify women in this cohort protected from placental malaria with high accuracy (though this should be validated in a second cohort). Our results suggest that multiple pathways exist that lead to the development of protective immunity against placental malaria. Next steps will include validating this set of features in other sample sets from different geographical settings and also determining whether these antibody features predict protection from adverse pregnancy outcomes such as low birth weight. The immune responses identified are relevant to further development of pregnancy malaria-specific therapies such as vaccines or monoclonal antibodies and possibly to the identification of protective antibody against malaria in non-pregnant populations. Multiple antibody functions and characteristics may be required to develop robust protective immunity to placental malaria.

Dataset used for *Figures 1–6* has been uploaded to Dryad: doi:10.5061/dryad.wpzgmsbkx.

## Acknowledgements

This work was supported by Project Grants from the National Health and Medical Research Council of Australia (NHMRC) to SJR, EHA, and AWC (APP1143946) and to PMH and BDW (GNT1145303) and a Program Grant (APP1092789) also from the NHMRC. AWC is supported by an NHMRC Career

Development Fellowship (APP1140509). AO-P is supported by scholarships from the University of Melbourne and the Miller Foundation. Work by EHA while visiting the University of Copenhagen was supported by a grant from the Australian Society for Parasitology. JAS is supported by a NHMRC Senior Research Fellowship (APP1104975). Sample collection was primarily supported by the Malaria in Pregnancy Consortium, through a grant from the Bill and Melinda Gates Foundation (46099). We thank Patrick Duffy and David Narum, Laboratory of Malaria Immunology and Vaccinology (LMIV) at NIAID, NIH, for provision of recombinant VAR2CSA proteins, Anne Corfitz, Centre for Medical Parasitology, Copenhagen, for technical assistance, and Isobel Walker, Graham Brown, Siddhartha Mahanty, and Louise Randall for constructive feedback on the work. We acknowledge Vanta Jameson and Alexis Gonzalez from the Melbourne Cytometry Platform (Melbourne Brain Centre node) for assistance with flow cytometry. The funders had no role in the study design, data collection, analysis, and interpretation of the data, or in the decision to submit the work for publication.

## Additional information

### Funding

| Funder | Grant reference number | Author |
|---|---|---|
| National Health and Medical Research Council | APP1143946 | Elizabeth H Aitken<br>Amy W Chung<br>Stephen J Rogerson |
| National Health and Medical Research Council | GNT1145303 | P Mark Hogarth<br>Bruce D Wines |
| National Health and Medical Research Council | APP1092789 | Stephen J Rogerson |
| National Health and Medical Research Council | APP1140509 | Amy W Chung |
| University of Melbourne | | Amaya Ortega-Pajares |
| Australian Society for Parasitology | | Elizabeth H Aitken |
| National Health and Medical Research Council | APP1104975 | Julie A Simpson |
| Bill and Melinda Gates Foundation | 46099 | Stephen J Rogerson |
| Miller Foundation Australia | | Amaya Ortega-Pajares |

The funders had no role in study design, data collection and interpretation, or the decision to submit the work for publication.

### Author contributions

Elizabeth H Aitken, Conceptualization, Data curation, Formal analysis, Supervision, Funding acquisition, Investigation, Visualization, Methodology, Writing - original draft, Project administration, Writing - review and editing; Timon Damelang, Amaya Ortega-Pajares, Formal analysis, Validation, Investigation, Visualization, Methodology, Writing - review and editing; Agersew Alemu, Validation, Investigation, Methodology; Wina Hasang, Supervision, Validation, Investigation, Methodology, Project administration; Saber Dini, Data curation, Formal analysis, Investigation, Visualization, Methodology, Writing - review and editing; Holger W Unger, Data curation, Writing - review and editing; Maria Ome-Kaius, Data curation; Morten A Nielsen, Resources, Methodology, Writing - review and editing; Ali Salanti, Joe Smith, P Mark Hogarth, Bruce D Wines, Resources, Writing - review and editing; Stephen Kent, Resources, Supervision, Writing - review and editing; Julie A Simpson, Formal analysis, Supervision, Methodology, Writing - review and editing; Amy W Chung, Conceptualization, Resources, Supervision, Funding acquisition, Methodology, Project administration, Writing - review and editing; Stephen J Rogerson, Conceptualization, Resources, Supervision, Funding acquisition, Methodology, Writing - original draft, Project administration, Writing - review and editing

## Author ORCIDs

Elizabeth H Aitken (iD) https://orcid.org/0000-0002-2677-6208
Timon Damelang (iD) http://orcid.org/0000-0002-6150-4435
Morten A Nielsen (iD) http://orcid.org/0000-0003-2668-4992
Joe Smith (iD) http://orcid.org/0000-0002-7915-6360
Julie A Simpson (iD) http://orcid.org/0000-0002-2660-2013
Stephen J Rogerson (iD) https://orcid.org/0000-0003-4287-1982

## Ethics

Clinical trial registration NCT01136850.
Human subjects: Collection and use of plasma samples from women in PNG was approved by the PNG Institute of Medical Research Institutional Review Board, the PNG Medical Research Advisory Council and the Melbourne Health Human Research Ethics Committee. All participants provided informed written consent. The use of blood products from donors in Melbourne for isolation of primary cells, culture of parasites and leukocytes and for use as negative controls was approved by the Melbourne Health Human Research Ethics committee and the University of Melbourne Human Research Ethics committee.

## Decision letter and Author response

Decision letter https://doi.org/10.7554/eLife.65776.sa1
Author response https://doi.org/10.7554/eLife.65776.sa2

# Additional files

## Supplementary files

• Supplementary file 1. Recombinant VAR2CSA DBL domain proteins used to measure antibody features.

• Supplementary file 2. Univariate analysis non-infected vs. placental malaria and non-placental infection vs. placental malaria.

• Supplementary file 3. Antibody feature code, name, and description.

• Supplementary file 4. Table 1: Association between selected antibody features and gravidity[a] in 77 women (women with placental malaria or non-placental infection at delivery) and Table 2: univariate analysis of selected antibody features placental malaria v non-placental infection, in women uninfected at enrollment.

• Transparent reporting form

• Reporting standard 1. Tripod checklist.

## Data availability

All antibody feature data has been deposited in Dryad.

The following dataset was generated:

| Author(s) | Year | Dataset title | Dataset URL | Database and Identifier |
|---|---|---|---|---|
| Aitken EH, Damelang T, Ortega-Pajares A, Alemu A, Hasang W, Dini S, Unger HW, Ome-Kaius M, Nielsen MA, Salanti A, Smith J, Kent SJ, Hogarth PM, Wines BD, Simpson JA, Chung AW, Rogerson SJ | 2021 | Antibody features towards VAR2CSA and CSA binding infected erythrocytes in a cohort of pregnant women from PNG | https://doi.org/10.5061/dryad.wpzgmsbkx | Dryad Digital Repository, 10.5061/dryad.wpzgmsbkx |

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

# Appendix 1

**Appendix 1—key resources table**

| Reagent type (species) or resource | Designation | Source or reference | Identifiers | Additional information |
|---|---|---|---|---|
| Strain, strain background (*Plasmodium falciparum*, CS2) | CS2 | *Chandrasiri et al., 2014* | | |
| Strain, strain background (*Plasmodium falciparum*, 3D7) | 3D7 | *Chandrasiri et al., 2014* | | Selected for chondroitin sulfate A (CSA) adhesion |
| Strain, strain background (*Plasmodium falciparum*, FCR3) | FCR3 | *Nielsen and Salanti, 2015* | | Selected for CSA adhesion (parent of CS2) |
| Strain, strain background (*Plasmodium falciparum*, NF54) | NF54 | *Nielsen and Salanti, 2015* | | Selected for CSA adhesion (parent of 3D7) |
| Biological sample (*Homo sapiens*) | Plasma | *Unger et al., 2015* | | |
| Biological sample (*Homo sapiens*) | Primary monocytes | This paper | | Freshly isolated cells (see Materials and methods – Primary leukocytes) |
| Biological sample (*Homo sapiens*) | Primary neutrophils | This paper | | Freshly isolated cells (see Materials and methods – Primary leukocytes) |
| Biological sample (*Homo sapiens*) | Primary NK cells | This paper | | Freshly isolated cells (see Materials and methods – Primary leukocytes) |
| Cell line (*Homo sapiens*) | THP1 | *Ataíde et al., 2010* | RRID: CVCL_0006 | Monocytic cell line |
| Peptide, recombinant protein | DBL1-7G8 | *Avril et al., 2011* | MV-1398 | Parasite line 7G8 |
| Peptide, recombinant protein | DBL2(ID1-ID2a)-FCR3 | *Doritchamou et al., 2016* | MV1942 | Parasite line FCR3 |
| Peptide, recombinant protein | DBL2(ID1-ID2a)-FCR3 | *Mordmüller et al., 2019* | | Parasite line FCR3 |
| Peptide, recombinant protein | DBL2-isolate | *Doritchamou et al., 2016* | MV 1940 | Parasite isolate 1010 |
| Peptide, recombinant protein | DBL3- FCR3 | *Nielsen et al., 2009* | MP1028 | Parasite line FCR3 |
| Peptide, recombinant protein | DBL3- 7G8 | *Avril et al., 2011* | MV-1914 | Parasite line 7G8 |
| Peptide, recombinant protein | DB4-FCR3 | *Fried et al., 2018* | MP2369 | Parasite line FCR3 |
| Peptide, recombinant protein | DBL4-isolate | *Doritchamou et al., 2016* | MV1700 | Parasite isolate I 0711 |
| Peptide, recombinant protein | DBL5-3D7 | *Avril et al., 2011* | 1218 | Parasite line 3D7 |
| Peptide, recombinant protein | DBL5-7G8 | *Avril et al., 2011* | 1269 | Parasite line 7G8 |
| Peptide, recombinant protein | DBL5-isolate | *Doritchamou et al., 2016* | MV 1749 | Parasite isolate I 0466 |
| Peptide, recombinant protein | DBL6-IT4 | *Avril et al., 2011* | MV-1137 | Parasite line IT4 |

*Continued on next page*

*Appendix 1—key resources table continued*

| Reagent type (species) or resource | Designation | Source or reference | Identifiers | Additional information |
|---|---|---|---|---|
| Peptide, recombinant protein | MSP-1 | *Barua et al., 2019* | | |
| Biological sample (*Plasmodium falciparum*) | Schizont extract | *Barua et al., 2019* | | |
| Antibody | Goat anti-human IgG (polyclonal) | Mabtech | 3820-4-250 | (1:2000) |
| Antibody | Mouse anti-human IgG-PE (polyclonal) | SouthernBiotech | 9040-09 RRID:AB_2796601 | (1:77) |
| Antibody | Mouse anti-human IgG1-PE (monoclonal) | SouthernBiotech | 9052-09 RRID:AB_2796621 | (1:77) |
| Antibody | Mouse anti-human IgG2-PE (monoclonal) | SouthernBiotech | 9070-09 RRID:AB_2796639 | (1:77) |
| Antibody | Mouse anti-human IgG3-PE (monoclonal) | SouthernBiotech | 9210-09 RRID:AB_2796701 | (1:77) |
| Antibody | Mouse anti-human IgG4-PE (monoclonal) | SouthernBiotech | 9200-09 RRID:AB_2796693 | (1:77) |
| Antibody | Mouse anti-human IgA1-PE (monoclonal) | SouthernBiotech | 9130-09 RRID:AB_2796656 | (1:77) |
| Antibody | Mouse anti-human IgA2-PE (monoclonal) | SouthernBiotech | 9140-09 RRID:AB_2796664 | (1:77) |
| Antibody | Mouse anti-human IgM-PE (monoclonal) | SouthernBiotech | 9020-09 RRID:AB_2796577 | (1:77) |
| Antibody | Rabbit anti-human IgG (polyclonal) | DAKO | A0425 | (1:100) |
| Antibody | Mouse anti-human IgG1 HP6069 (monoclonal) | Merck Millipore | 411451 | (1:50) |
| Antibody | Mouse anti-human IgG2 HP6002 (monoclonal) | Merck Millipore | MAB1308 | (1:50) |
| Antibody | Mouse anti-human IgG3 HP6050 (monoclonal) | Sigma | I7260.2ml | (1:50) |

*Continued on next page*

*Appendix 1—key resources table continued*

| Reagent type (species) or resource | Designation | Source or reference | Identifiers | Additional information |
|---|---|---|---|---|
| Antibody | Mouse anti-human IgG4 HP6023 (monoclonal) | Merck Millipore | MAB1312-K | (1:50) |
| Antibody | Goat anti-mouse IgG AlexaFluor 647 (polyclonal) | Life Technologies | A-21235 RRID:AB_2535804 | (1:500) |
| Antibody | Donkey anti-rabbit 647 (polyclonal) | Life Technologies | A-31573 RRID:AB_2536183 | (1:500) |
| Peptide, recombinant protein | C1q | MP Biomedicals | 80295-33-6 | 1.3 µg/ml |
| Peptide, recombinant protein | FcγRI | R&D Systems | 1257-FC | |
| Peptide, recombinant protein | FcγRIIa | *Wines et al., 2016* | | |
| Peptide, recombinant protein | FcγRIIIa | *Wines et al., 2016* | | |
| Peptide, recombinant protein | FcγRIIIb | R&D Systems | 1875 CD | |
| Antibody | Mouse anti-human CD16-Brilliant violet 605 (monoclonal) | BD | 563172 | (1:50) |
| Antibody | Mouse anti-human CD56-Brilliant ultraviolet 737 (monoclonal) | BD | 347344 | (1:800) |
| Antibody | Mouse anti-human CD3-peridinin-chlorophyll-protein (monoclonal) | BD | 552127 | (1:200) |
| Antibody | Mouse anti-human IFNγ-PE (monoclonal) | BD | 554701 | (1:200) |
| Antibody | Mouse anti-human TNFα-BV-785 (monoclonal) | BioLegend | 502947 RRID:AB_2565857 | (1:200) |
| Software, algorithm | R software, caret package | R software *Kuhn, 2008* | | |
| Software, algorithm | R software, glmnet package | R software *Zou and Hastie, 2005* | | |
| Software, algorithm | R software, PLS package | R software *Mevik and Wehrens, 2007* | | |

*Continued on next page*

*Appendix 1—key resources table continued*

| Reagent type (species) or resource | Designation | Source or reference | Identifiers | Additional information |
|---|---|---|---|---|
| Software, algorithm | R software, mixOmics package | R software *Rohart et al., 2017* | | |
| Software, algorithm | R software, qgraph package | R software *Epskamp et al., 2012* | | |
| Commercial assay or kit | EasySep Direct Human Neutrophil Isolation Kit | STEMCELL Technologies | 19666 | |
| Commercial assay or kit | RosetteSep Human Monocyte Enrichment Cocktail | STEMCELL Technologies | 15068 | |
| Commercial assay or kit | RosetteSep Human NK Enrichment Cocktail | STEMCELL Technologies | 15065 | |
| Commercial assay or kit | Melon Gel purification kits | Thermo Fisher Scientific | 45212 | |
| Commercial assay or kit | EZ-link Sulfo NHS-LC-Biotin kit | Thermo Fisher Scientific | 24520 | |
| Other | Bio-Plex magnetic carboxylated microspheres | Bio-Rad | #MC100xx-01 | |
| Other | Streptavidin-PE | SouthernBiotech | 7105-09 | |

