## [Decision Letter]

**Acceptance summary:**

This is the very first study to apply systems serology approaches to study antibody responses directed to specific Plasmodium (malaria) antigens and the results demonstrate that these responses correlate with protection from placental malaria. The analyses of samples from approximately 200 women revealed that these antibody features do discriminate between women with placental malaria at delivery and women with malaria detectable in the peripheral blood, but not in placenta. These findings support the effort to develop vaccines against the certain protein antigens of placental malaria parasites.

**Decision letter after peer review:**

Thank you for submitting your article "Identifying women protected from placental malaria using systems serology" for consideration by *eLife*. Your article has been reviewed by 3 peer reviewers, and the evaluation has been overseen by a Reviewing Editor and Dominique Soldati-Favre as the Senior Editor. The following individual involved in review of your submission has agreed to reveal their identity: Sidhartha Chaudhury (Reviewer #3).

The Editors and reviewers have discussed your revision plan with one another, and the Reviewing Editor has drafted this to help you prepare a revised submission.

Essential revisions:

1) Your responses to the revisions as proposed by the Reviewers have, in general, been considered as sufficient for addressing their concerns.

2) However, apart from the specific critical points as indicated in the initial decision letter and which are included here once again, the revised manuscript should clearly state your definition of the primary group used to analyze correlates of protection – the "non-placental infection at delivery" group. At present, it appears to be defined as "women with *P. falciparum* infection by light microscopy and/or PCR in the peripheral blood but with no IEs visible in the placenta by histology". This response ("Line 85") leaves open whether this definition means "no IEs" or "no mature IEs" by histology. Thus, it seems that a full clear definition is missing in the manuscript, and it is not clear from your response what the planned revision includes. In addition, please indicate in Table 1 how many of the 27 women in this group were positive by microscopy versus PCR of their peripheral blood.

*Reviewer #1:*

The authors have conducted a systems serology study of the relationship between antibody responses to VAR2CSA and protection from placental malaria (PM) in pregnant women from Papua New Guinea.

Based on initial analysis of 130 variables and samples from 127 women (15,340 data points) and subsequent analysis of 169 variables from 77 women (12,906 data points), they identified a list of six selected serological features that discriminated with 86% accuracy between women with placental parasitaemia at delivery and women with peripheral but not placental parasitaemia at the same time point. It is an important conclusion from the work performed that it underscores the likelihood that multiple pathways to clinical protection from PM exists – a conclusion that can safely be extended to other clinical categories of malaria – and probably even to infectious diseases in general. While this may seem obvious to some, it is nevertheless largely ignored in almost all existing reports. In particular, current efforts to develop vaccines against PM are completely focussed on inhibition of adhesion of infected erythrocytes (IEs), completely ignoring the antibody effector functions that rely on opsonization of IEs for cell-mediated destruction, and which the current analysis suggest are important.

The manuscript is very well written, based on careful and in-depth analysis, and constitutes a valuable addition to the existing literature.

While I am not able to ascertain the finer points of the advanced statistical analyses conducted, I have no reason to suspect that anything is amiss.

I have only a couple of specific comments:

P. 14 ("ADNP of IE… contributed to protection"): I think it would be more correct to write "ADNP of IE… correlated with placental infection status" as no causal relationship was established.

P. 15: The authors list weaknesses of their study, which is commendable. However, they fail to mention what I consider a more prominent weakness than those listed, namely the absence of a test of their six-features-list on an independent data/sample set. While such a test would be optimal, at least it should be mentioned as a weakness if it is not done.

*Reviewer #2:*

The authors are the first to use systems serology to examine antibody responses associated with protection from placental malaria. They determine that functional antibodies against CSA-binding IE are associated with protection from placental malaria, in general confirming findings from earlier studies. While the consistency with earlier studies engenders confidence in their findings, it does lessen the novelty. Important aspects of the clinical study design and study cohort need better description and justification. The study would be strengthened considerably if the top features identified here were tested for their predictive capacity in a second confirmatory cohort, albeit the fact that these associations have previously been made in other studies give general confidence in the findings. Protective immunity against placental malaria increases with gravidity, and therefore the authors should report how the protective features that they identify are related to gravidity.

Abstract concludes "inhibition of placental binding and/or opsonic phagocytosis of infected erythrocytes" identify women who are protected from placental malaria. This is consistent with earlier studies using non-systems approaches that previously identified these associations.

Line 85 "some pregnant women with peripheral blood infection remain free of placental malaria" – while such cases have been sporadically described, the "placental malaria-negative infection" phenotype is an uncommon phenomenon in studies that carefully document placental and peripheral Pf infections, and requires better definition here. Who has used such a study design before? "Placental malaria-negative" can mean that mature parasites do not sequester in the placenta, but does not necessarily mean that all parasites are absent from the placenta. Practically speaking, how could peripheral ring stage parasites be excluded from circulating through the intervillous spaces at the same density they are found in the peripheral blood? Related to this, the authors should provide detailed information on the pregnancy outcomes for the women in the different groups: especially Pf quantification in peripheral blood and placental blood by microscopy and by PCR, etc, but also BW, gestational age. It may be that the "PM-negative infection" phenotype is related to highly sensitive detection of sparse Pf in peripheral blood by PCR versus less sensitive detection of sparse Pf in placental blood using Giemsa-stained histology (see similar comment below).

Line 108 "Five of the six leading antibody features we found were related to inhibition of placental sequestration and/or opsonisation for phagocytic clearance" Can the authors comment whether the same antibodies might be mediating both effects, or do they surmise they are unlikely to overlap?

Line 112 Please state briefly but clearly here how the samples/women were selected for inclusion in the cohort (in addition to the more detailed description in the Methods).

Line 115 "All women resided in heavily malaria exposed areas" is not consistent with the evidence provided in Table 1 that only 14/127 women had Pf detected by PCR at their baseline visit. The authors should explain this. Had these women previously received IPTp, or was this their baseline sample collected before first IPTp?

Line 119 "no *P. falciparum* parasites in the placenta but who had peripheral blood parasitemia by PCR and/or light microscopy" This wording and the Methods section indicate peripheral blood was tested for Pf by Giemsa and PCR, but placental blood was tested by Giemsa but not PCR. This is not a fair comparison. What is the sensitivity of their PCR versus their Giemsa stained blood or tissue?

Line 142 "using samples from the 127 women and measuring 13 antibody features directed at ten DBL domains, 15,340 out of 16,510 possible data points were acquired". This is an impressive dataset to examine correlates of protection.

All pregnant women in this study have received IPTp and this can affect the PM outcome at delivery. Looks like this has not been really considered in the interpretation of the data, particularly in view of the fact that two types of IPTp combinations were used, and readers are not informed of their potentially different efficacies.

Among the identified antibody features in this study it appeared that none of the activities targeted more than one variant of protein or parasite. Can the authors discuss this lack of strain-transcending pattern in these activities? A similar pattern of parasite variant dependent activity of PM antibody has previously been described by Doritchamou et al. IAI 2019. The authors should discuss their findings in that context and also comment whether the six antibody features identified in this study might vary in populations from different geographical areas.

Lines 214 – 215 "… the six features identified by elastic net were spread throughout the correlation network and were not well correlated with each other (Figure 4 and Supplementary Figure 3)". Based on this statement it is confusing when the authors conclude that the six features fall into two major functional groups. In particular the inverse correlation between IgG3.DBL2(ID1-ID2a).FCR3 with CSA.Binding.Inhibition.FCR3 is inconsistent with Figure 6 suggesting that these represent related responses.

The authors indicated that 17 samples were Pf positive at enrolment. What is the impact of early infection on these identified antibody features in relation to protection against PM and to antibody responses?

*Reviewer #3:*

The authors hypothesize that protection from placental malaria requires specific humoral immunity to malaria protein VAR2CSA. The authors obtained plasma samples from 127 pregnant women and grouped them into three categories based on malaria diagnosis at delivery: non-infected at delivery, placental malaria at delivery, and non-placental malaria at delivery. They use systems serology to measure 169 variables related to antibody binding, isotype, and functional activity and used machine learning to identify 6 variables that could accurately distinguish between placental and non-placental malaria cases. They delve into these 6 variables to provide some insight into the mechanisms of protection against placental malaria.

Strengths: The cohorts in this study are large and robust, with 50 placental and 27 non-placental cases with similar demographic profiles. They assess a wide range of serology measures for binding, isotype, isotype subclass, and functional responses against both infected erythrocyte and recombinant antigens. They find significant group-level differences in serology measures between placental and non-placental malaria groups suggesting that there is great potential in this data set to identify correlates of immunity and potential mechanisms of protection.

Weaknesses: They used a elastic network-regularized linear regression method to identify from 169 variables, the top variables that distinguish placental and non-placental malaria cases, and then use a Partial Least Square Discriminant Analysis method to assess how accurately a subset of these variables could be used to predict placental malaria susceptibility. While this overall approach is reasonable, it contains a substantial risk of overfitting that is not properly assessed in the study. Overfit models produce seemingly highly accurate predictions that result from incidental/random variation in the data rather than biologically significant differences, and jeopardize the scientific conclusions of the study.

Furthermore, the authors assume that differences between the placental malaria and non-placental malaria groups reflect correlates of protection, thus justifying an exploration into mechanisms of protection. However, these two groups could reflect differences in malaria exposure, despite demographic similarities in age/residence. If correlates of protection cannot be distinguished from correlates of exposure, this limits what can be inferred about mechanisms of protection.

Appraisal: The authors conclusions rest substantially on the six variables identified through machine learning. Ultimately the study conclusions cannot be supported by the data without demonstrating that these six variables reflect biologically significant differences and do not simply arise from analysis choices and model overfitting.

1) It is important to demonstrate that malaria exposure is similar in the placental and non-placental malaria groups because a key assumption made here is that a comparison between these groups identifies correlates of protection, and not simply correlates of exposure. Indeed, the authors find that the non-placental malaria group had almost double the PCR positive rate of the placental malaria group at enrolment, suggesting it is possible they had higher exposure, despite similar demographic profiles.

One way to address this would be to take serological markers in the data that others in literature have shown to be linked to exposure, and show that there is no significant differences between the three groups with respect to these serological markers of exposure.

2) Much of the conclusions of this study rests on the outcome of the machine learning analysis, so it is critical that each step of this analysis is properly justified.

In the elastic network-regularized logistic regression (ENLR), more detail is needed to explain how correlated variables are handled. For example, why was α set to 0.50, as opposed to any other value between 0 and 1? To what degree are the findings that a small number of non-correlated variables predictive of placental malaria susceptibility simply a reflection of the analysis approach rather than the underlying biology?

3) Only three variables out of the top 20 identified by logistic regression have odds-ratios whose lower bound IQR exceeds 1.0 (Figure 2). Does this suggest the observed effects are not significant? How was it decided that 20 variables would be selected for further study?

4) In the Partial Least Squares Discriminant Analysis (PLSDA), was any cross-validation done? Or was the model trained on the entire data set? PLSDA can be highly susceptible to overfitting, and cross-validation should be done to assess prediction accuracy.

Also, given the imbalance in the data set (2:1 ratio of placental to non-placental malaria cases), other measures of accuracy should be considered, such as sensitivity, specificity, Cohen's kappa value, and/or ROC AUC.

5) An assessment of the this approach (ENLR + PLSDA) in terms of its risk for overfitting must be done. Guided feature reduction (ENLR) combined with PLSDA carries a high risk of overfitting. A comparison with 6 random variables in PLSDA is not sufficient to assess overfitting because it doesn't account for the degree to which guided feature reduction (ENLR) step contributes to overfitting.

Authors should consider doing a permutation test, where the data set is randomized with respect to outcome (placental vs. non-placental malaria), and then the entire ENLR + PLSDA method is carried out to determine prediction accuracy. This can be carried out for N different permutations (for example N=100 times) to determine the average prediction accuracy for this randomized outcome. Because the outcome is randomized, it is decoupled from the immune data, and thus acts as a 'negative control' for machine learning and it should fail to predict the outcome with any accuracy greater than chance. The degree to which it does 'accurately' predict this randomized outcome is a reflection of the overfitting of the model.

---

## [Author Response]

Essential revisions:The revised manuscript should clearly state your definition of the primary group used to analyze correlates of protection – the "non-placental infection at delivery" group. At present, it appears to be defined as "women with *P. falciparum* infection by light microscopy and/or PCR in the peripheral blood but with no IEs visible in the placenta by histology". This response ("Line 85") leaves open whether this definition means "no IEs" or "no mature IEs" by histology. Thus, it seems that a full clear definition is missing in the manuscript, and it is not clear from your response what the planned revision includes. In addition, please indicate in Table 1 how many of the 27 women in this group were positive by microscopy versus PCR of their peripheral blood.

We have added further clarification to the text under Methods, lines 148 onwards and Results, Cohort description, starting at line 467, to indicate that placental malaria was defined as no IEs visible by histology (JN Bulmer, Histopathology 1993; PR Walter et al., Am J Path 1982). As those authors observed, histology is most sensitive at detecting mature trophozoites and schizonts, in part because they actively accumulate in the intervillous space, and in part because they are both larger and contain visible hemozoin. We did record presence of any IEs on histology examination irrespective of stage as placental malaria.

We have expanded Table 1 to add the results of peripheral blood PCR and microscopy. See also response to reviewer 2.

Reviewer #1:The authors have conducted a systems serology study of the relationship between antibody responses to VAR2CSA and protection from placental malaria (PM) in pregnant women from Papua New Guinea.Based on initial analysis of 130 variables and samples from 127 women (15,340 data points) and subsequent analysis of 169 variables from 77 women (12,906 data points), they identified a list of six selected serological features that discriminated with 86% accuracy between women with placental parasitaemia at delivery and women with peripheral but not placental parasitaemia at the same time point. It is an important conclusion from the work performed that it underscores the likelihood that multiple pathways to clinical protection from PM exists – a conclusion that can safely be extended to other clinical categories of malaria – and probably even to infectious diseases in general. While this may seem obvious to some, it is nevertheless largely ignored in almost all existing reports. In particular, current efforts to develop vaccines against PM are completely focussed on inhibition of adhesion of infected erythrocytes (IEs), completely ignoring the antibody effector functions that rely on opsonization of IEs for cell-mediated destruction, and which the current analysis suggest are important.The manuscript is very well written, based on careful and in-depth analysis, and constitutes a valuable addition to the existing literature.While I am not able to ascertain the finer points of the advanced statistical analyses conducted, I have no reason to suspect that anything is amiss.I have only a couple of specific comments:P. 14 ("ADNP of IE… contributed to protection"): I think it would be more correct to write "ADNP of IE… correlated with placental infection status" as no causal relationship was established.

We agree and have revised the sentence as suggested.

Line 678-680 “*ADNP of IE, which has not previously been systematically studied in malaria immunity also correlated with placental infection status”P. 15: The authors list weaknesses of their study, which is commendable. However, they fail to mention what I consider a more prominent weakness than those listed, namely the absence of a test of their six-features-list on an independent data/sample set. While such a test would be optimal, at least it should be mentioned as a weakness if it is not done.*

We have added the following to the section on study weaknesses:

Line 760-2 “*Demonstration that our selected variables correlate with lack of placental malaria in other cohorts would further strengthen the findings*.”

Reviewer #2:The authors are the first to use systems serology to examine antibody responses associated with protection from placental malaria. They determine that functional antibodies against CSA-binding IE are associated with protection from placental malaria, in general confirming findings from earlier studies. While the consistency with earlier studies engenders confidence in their findings, it does lessen the novelty. Important aspects of the clinical study design and study cohort need better description and justification. The study would be strengthened considerably if the top features identified here were tested for their predictive capacity in a second confirmatory cohort, albeit the fact that these associations have previously been made in other studies give general confidence in the findings. Protective immunity against placental malaria increases with gravidity, and therefore the authors should report how the protective features that they identify are related to gravidity.

We thank the reviewer for their helpful comments. We have added a comment to the discussion under study weaknesses regarding the need to confirm our findings in further cohorts (Lines 760-2, 765-767), see also response to Reviewer 1. Our plans to examine responses in a second confirmatory cohort have been severely disrupted by COVID-19, especially in regard to sample shipments which have not been possible for the past 12 months.

Regarding associations of protective features and gravidity, we have conducted a univariate analysis using linear regression looking at how the antibody levels vary with gravidity in our cohort and have added this to the paper in the form of Supplementary File 4-table 1.

In brief: Antibody feature CSA.Binding.Inhibition.FCR3 was positively associated with gravidity, and there was weak evidence that THP1.Phago.CS2 and Neutrophil.Phago.CS2 were also positively associated with gravidity. There was no evidence that IgG3.DBL2(ID1-ID2a).FCR3, IgA2.DBL2.1010 and IgG3.3D7 were associated with gravidity in this cohort. We now discuss these observations in the manuscript (lines 573-585).

Abstract concludes "inhibition of placental binding and/or opsonic phagocytosis of infected erythrocytes" identify women who are protected from placental malaria. This is consistent with earlier studies using non-systems approaches that previously identified these associations.

We made small edits to the abstract to reflect that selected protective responses include features previously described:

Line 50-51 *“Selected features included previously described associations with inhibition of placental binding and/or opsonic phagocytosis of infected erythrocytes.”*

Line 85 "some pregnant women with peripheral blood infection remain free of placental malaria" – while such cases have been sporadically described, the "placental malaria-negative infection" phenotype is an uncommon phenomenon in studies that carefully document placental and peripheral Pf infections, and requires better definition here. Who has used such a study design before? "Placental malaria-negative" can mean that mature parasites do not sequester in the placenta, but does not necessarily mean that all parasites are absent from the placenta. Practically speaking, how could peripheral ring stage parasites be excluded from circulating through the intervillous spaces at the same density they are found in the peripheral blood? Related to this, the authors should provide detailed information on the pregnancy outcomes for the women in the different groups: especially Pf quantification in peripheral blood and placental blood by microscopy and by PCR, etc, but also BW, gestational age. It may be that the "PM-negative infection" phenotype is related to highly sensitive detection of sparse Pf in peripheral blood by PCR versus less sensitive detection of sparse Pf in placental blood using Giemsa-stained histology (see similar comment below).

We agree with the reviewer that women with peripheral blood infection without evidence of placental sequestration form a relatively rare group, and we only identified 27 in our cohort. They are analogous to infections that are found in non-pregnant women in malarious areas. When women conceive, some of these infections progress to higher density, especially in younger, less immune women (Ndam et al. Clin Infect Dis 2018; Hounkonnou et al. Clin Infect Dis 2020), likely due to emergence of variants than can sequester in the placenta and avoid existing antibody immunity.

Previous studies (reviewed by Cutts et al., BMC Med 2020) highlight the difficulty of distinguishing effects of exposure from those of immunity using the more common comparison of women with and without placental malaria, which was also not informative in our hands (Figure 1 a), compared to our novel inclusion of this additional group (Figure 1 b).

The reason we (and many other studies) use detection of IEs in the placenta by histology to define placental malaria, is because many studies show that these infections are associated with worse pregnancy outcomes such as prematurity, low birth weight or maternal anemia (Menendez C et al., J Infect Dis 2000; Rogerson SJ et al., Am J Trop Med Hyg 2003). Identifying women whose infection has not progressed to placental malaria, and the antibody responses that might limit that progression by inhibiting placental sequestration, is our specific aim.

The circulating ring stage parasites will pass through the placenta although they do not sequester there, and placental blood PCR did detect them in some women (added to Table 1). These circulating rings are understood not to have any deleterious effects on the placenta, and submicroscopic infections, including in placental blood, appear to be only weakly associated with adverse birth outcomes, unlike active placental infection on histology.

We have included further information as requested on the pregnancy outcomes, including BW, hemoglobin and gestational age at delivery and peripheral and placental PCR in Table 1. Original groupings of the cohorts were not done using placental PCR, however subsequent analysis revealed that one individual classified as non-infected (which was based on PCR and light microscopy of peripheral blood as well as placental histology) was actually PCR positive in the placental blood. As our analysis identifying protective features did not include the non-infected cohorts, this individual’s results do not affect the conclusions in this paper.

Line 108 "Five of the six leading antibody features we found were related to inhibition of placental sequestration and/or opsonisation for phagocytic clearance" Can the authors comment whether the same antibodies might be mediating both effects, or do they surmise they are unlikely to overlap?

The diversity of responses in “protected” women in Figure 5G suggests that these often do not overlap, or the overlap is heterogenous in different women. However, determining whether individual antibodies have overlapping functions would ideally require monoclonals and therefore we cannot confidently comment on whether the same antibody might be able to mediate both functions.

Line 112 Please state briefly but clearly here how the samples/women were selected for inclusion in the cohort (in addition to the more detailed description in the Methods).

As suggested, in addition to the detailed description in the methods, we have briefly described the selection of the cohort in the Results section (which now follows methods, as requested by the Editors), see below:

LINE 467 onwards: “Women were selected based on *P. falciparum* infection status at delivery […] The three groups were frequency matched for primigravidity, IPTp regime receipt, bed net use, rural residency and age”

Line 115 "All women resided in heavily malaria exposed areas" is not consistent with the evidence provided in Table 1 that only 14/127 women had Pf detected by PCR at their baseline visit. The authors should explain this. Had these women previously received IPTp, or was this their baseline sample collected before first IPTp?

Line 466: We have removed the word “heavily”, given the relatively low parasite prevalence. In an earlier study at one of our nine sites we found 64% of women to have current or past placental malaria on histology and 34% and 49% to have *P. falciparum* at first antenatal visit by microscopy and PCR respectively (Stanisic et al. Trans Roy Soc Trop Med Hyg 2015), and transmission in the area had only recently declined (Ome-Kaius et al. BMC Med 2019).

All women were recruited at first antenatal visit before receiving any IPTp. We have clarified this under “Human Subjects” in the methods in line 143-4.

“Peripheral blood collected at 14-26 gestation weeks, before initiation of IPTp, was used in the antibody assays.”Line 119 "no *P. falciparum* parasites in the placenta but who had peripheral blood parasitemia by PCR and/or light microscopy" This wording and the Methods section indicate peripheral blood was tested for Pf by Giemsa and PCR, but placental blood was tested by Giemsa but not PCR. This is not a fair comparison. What is the sensitivity of their PCR versus their Giemsa stained blood or tissue?

See our earlier response to line 85. We specifically grouped women with malaria infection into those with and without placental malaria on histology, as classically defined (Walter Am J Path 1982; Bulmer, Histopathology 1993). Our question was, are there antibody responses that might explain why some infected women have not progressed to placental malaria, when others have. Antibody responses that prevent infection crossing a threshold (to symptomatic infection, to placental infection or to severe disease) are more relevant to blood stage immunity than antibodies that simply prevent infection, as may be sought for pre-erythrocytic infection.

Line 142 "using samples from the 127 women and measuring 13 antibody features directed at ten DBL domains, 15,340 out of 16,510 possible data points were acquired". This is an impressive dataset to examine correlates of protection.All pregnant women in this study have received IPTp and this can affect the PM outcome at delivery. Looks like this has not been really considered in the interpretation of the data, particularly in view of the fact that two types of IPTp combinations were used, and readers are not informed of their potentially different efficacies.

We added a statement to the results highlighting the different efficacy of SPAZ and SPCQ against placental malaria (Line 476-7). As shown in Table 1, a slightly higher proportion of women in the non-placental malaria group received the less efficacious SPCQ.

“Sulfadoxine pyrimethamine (SP) and azithromycin was more efficacious against placental malaria than SP-chloroquine(Unger et al., 2015).”

Among the identified antibody features in this study it appeared that none of the activities targeted more than one variant of protein or parasite. Can the authors discuss this lack of strain-transcending pattern in these activities? A similar pattern of parasite variant dependent activity of PM antibody has previously been described by Doritchamou et al. IAI 2019. The authors should discuss their findings in that context and also comment whether the six antibody features identified in this study might vary in populations from different geographical areas.

As suggested we have discussed our findings in relation to those described by Doritchamou et al. IAI 2019

Line 726 onwards: *None of the antibody features targeted more than one variant of the protein or parasite strain. One possible explanation is that epitopes recognised by antibodies mediating each function may vary between strains (as suggested by Doritchamou et al., 2019). This could in turn mean that protective functional features could vary with different populations (as there is variation in global distribution of the different VAR2CSA domain clades (Benavente et al., 2018)). Future studies should ideally include multiple VAR2CSA clades, and study diverse populations.*

Lines 214 – 215 "… the six features identified by elastic net were spread throughout the correlation network and were not well correlated with each other (Figure 4 and Supplementary Figure 3)". Based on this statement it is confusing when the authors conclude that the six features fall into two major functional groups. In particular the inverse correlation between IgG3.DBL2(ID1-ID2a).FCR3 with CSA.Binding.Inhibition.FCR3 is inconsistent with Figure 6 suggesting that these represent related responses.

The broad functional mechanisms of antibody can fall into similar groups, even when the development of responses to different members of each group is only weakly correlated. As the reviewer notes, there was no correlation between IgG3.DBL2(ID1-ID2a).FCR3 with CSA.Binding.Inhibition in Figure 4 and a weak, non-significant negative correlation between them (Figure 4—figure supplement 1). Although binding inhibition is likely to be primarily mediated by the more-abundant IgG1, we included IgG3 to DBL2 in both Figure 6A and Figure 6B because its principal functional role could fall in either group, as total IgG antibody to this domain is known to block adhesion.

The authors indicated that 17 samples were Pf positive at enrolment. What is the impact of early infection on these identified antibody features in relation to protection against PM and to antibody responses?

We thank the reviewer for asking this important question, as it has improved the quality of the manuscript. We have now conducted a sub-analysis comparing the identified antibody features at enrolment in the non-placental infection cohort with those in the placental malaria cohort, excluding women who were infected (by light microscopy and/or PCR) at enrolment. In this sub-analysis we saw strong evidence that 4 of the 6 features were higher in the non-placental infection cohort (P≤0.013) and weak evidence that the other two features were higher in the non-placental infection cohort (≤0.064). The maintenance of the protective associations in the smaller, non-infected cohort suggests to us that the protective association seen in these variables was not greatly affected by the presence of the positive samples.

These data are included in the manuscript in Supplementary File 4 – table 2. These results are discussed in lines 577 onwards.

Reviewer #3:1) It is important to demonstrate that malaria exposure is similar in the placental and non-placental malaria groups because a key assumption made here is that a comparison between these groups identifies correlates of protection, and not simply correlates of exposure. Indeed, the authors find that the non-placental malaria group had almost double the PCR positive rate of the placental malaria group at enrolment, suggesting it is possible they had higher exposure, despite similar demographic profiles.One way to address this would be to take serological markers in the data that others in literature have shown to be linked to exposure, and show that there is no significant differences between the three groups with respect to these serological markers of exposure.

Thank you for this suggestion. We measured IgG levels by ELISA to non-pregnancy specific malaria antigens merozoite surface protein-1 and schizont extract which we have previously shown to be strongly associated with exposure (Barua 2019 Malaria Journal) and have done univariate analyses comparing non-infected v placental malaria and non-placental infection v placental malaria. There was no association with absence of placental infection for IgG towards either antigen, suggesting that the features we have identified are not simply correlates of exposure.

We have included this in the Results section (line 510 onwards) and in Supplementary File 2.

2) Much of the conclusions of this study rests on the outcome of the machine learning analysis, so it is critical that each step of this analysis is properly justified.In the elastic network-regularized logistic regression (ENLR), more detail is needed to explain how correlated variables are handled. For example, why was α set to 0.50, as opposed to any other value between 0 and 1? To what degree are the findings that a small number of non-correlated variables predictive of placental malaria susceptibility simply a reflection of the analysis approach rather than the underlying biology?

To address the dependence of the results on α, we have now computed the selection frequencies across all α values {0,0.25,0.5,0.75,1} instead of only α= 0.5, using the resampling of ENLR explained in Section Identification of Key Antibody Features of the previous manuscript. The results showed that the top six variables do not change (please see Figure 2—figure supplement 1). We have now modified the revised manuscript to address this point.

Line 411 onwards now states:

Identification of Key Antibody Features

“The parameter α, which alters the nature of penalization between ridge regression and LASSO, was set to 0.5 to achieve a balance between sparsity and group selection; the final selected antibody features did not alter when α was varied over [0, 1], as detailed below. […] The resampling process was repeated where α was also tuned over the set {0,0.25,0.5,0.75,1} (instead of only α=0.5) in addition to λ and the results showed that the final top frequently selected variables do not change. (Figure 2—figure supplement 1)”

3) Only three variables out of the top 20 identified by logistic regression have odds-ratios whose lower bound IQR exceeds 1.0 (Figure 2). Does this suggest the observed effects are not significant? How was it decided that 20 variables would be selected for further study?

Due to the penalisation in the elastic net, the unselected variables are assigned a coefficient of zero (i.e. odds ratio = 1). Here we are repeating ENLR, resampling from the dataset 50,000 times, and therefore, the estimated coefficient for every variable is zero in a certain proportion of resampled datasets (even though a variable may frequently be selected in a high proportion of the resampled datasets). This pushes the lower bounds of the IQRs towards zero, hence lower bound of odds ratio will be very close to 1 for many of the variables. Therefore, taking the penalisation into consideration, this does not necessarily suggest that the effects are insignificant for those variables. Figure 2 presented the top 20 variables for ease of illustration.

4) In the Partial Least Squares Discriminant Analysis (PLSDA), was any cross-validation done? Or was the model trained on the entire data set? PLSDA can be highly susceptible to overfitting, and cross-validation should be done to assess prediction accuracy.

Yes, 500 repeats of 10-fold cross-validation was performed and the reported performance measures were computed from the cross-validation. This is now added to the “Identification of Key Antibody Features” section and the caption of Figure 3 is edited for clarity.

Identification of Key Antibody Features (line 440)

“…; 500 repeats of 10-fold cross validation were performed to compute the accuracies.”

Figure Legend 3

“… (500 repeats of 10-fold cross-validation were performed to estimate accuracy for each model)”Also, given the imbalance in the data set (2:1 ratio of placental to non-placental malaria cases), other measures of accuracy should be considered, such as sensitivity, specificity, Cohen's kappa value, and/or ROC AUC.

We have now changed figures 3A and 3B and the text (line 555 onwards) by presenting AUROC as the performance measure, as the reviewer suggests.

5) An assessment of the this approach (ENLR + PLSDA) in terms of its risk for overfitting must be done. Guided feature reduction (ENLR) combined with PLSDA carries a high risk of overfitting. A comparison with 6 random variables in PLSDA is not sufficient to assess overfitting because it doesn't account for the degree to which guided feature reduction (ENLR) step contributes to overfitting.Authors should consider doing a permutation test, where the data set is randomized with respect to outcome (placental vs. non-placental malaria), and then the entire ENLR + PLSDA method is carried out to determine prediction accuracy. This can be carried out for N different permutations (for example N=100 times) to determine the average prediction accuracy for this randomized outcome. Because the outcome is randomized, it is decoupled from the immune data, and thus acts as a 'negative control' for machine learning and it should fail to predict the outcome with any accuracy greater than chance. The degree to which it does 'accurately' predict this randomized outcome is a reflection of the overfitting of the model.

We thank the reviewer for raising this important point and agree that the estimated performances are probably slightly optimistic and the accuracy may slightly decrease if this method is applied on external validation data. We performed the analysis suggested by the reviewer, i.e. repeating the whole ENLR + PLSDA process for 100 datasets whose group labels (PM and NPM) were randomly permuted. For each of the 100 datasets, 500 repeats of 10-fold cross validation were performed on the EN and the top six frequently selected variables were fed into the PLSDA, on which 5 repeats of 10-fold cross-validation were performed to compute the AUROC. The AUROCs of the 100 datasets were aggregated to produce the boxplot in Figure 3B.

The results (Figure 3B) show that the AUROC for our six selected variables is still significantly higher than that for the permutation test.

Line 441 onwards now state:

Identification of Key Antibody Features

“The statistical significance of the results was assessed by comparing the performance of the model with that of two random permutation tests (null cases): 1) the PLSDA model was fitted to six randomly selected antibody features and the performance was computed for 500 repeats of 10-fold cross-validation resampling; 2) 100 datasets were generated by randomly permuting the group labels (PM and NPM) and the same analysis performed for the original data set (i.e. building PLSDA models using the top six frequently selected antibody features found by resampling of elastic net) was repeated for each dataset.”